# Aligning Path-based Link Prediction with Human Understanding of Valid Reasoning

## Abstract

Path-based link prediction methods reconstruct missing links between two vertices of a knowledge graph. They reconstruct a missing link by finding a path through the knowledge graph connecting both vertices. The path is the reasoning of the link prediction method. However, path-based link prediction methods are vulnerable to *Clever Hans* biases. They learn invalid reasoning patterns if these patterns are dominant and generalize well to the training and validation set. As a result, performance drops when evaluated on the real-world distribution. Whether reasoning is valid, that is, semantically relevant to the missing link and not merely factually correct, depends on the semantic concept underlying the link, which is mostly accessible through human knowledge. The paper's approach makes the understanding of a human oracle of valid reasoning accessible while learning to predict missing links. This paper proposes the path-based link prediction method *LiEr*. *LiEr* learns human-oracle-aligned reasoning within the knowledge graph domain from preference-based human feedback. The paper demonstrates that *LiEr*'s prediction capability is on par with other state-of-the-art link prediction methods, while its reasoning aligns more closely with human-specified notions of valid reasoning across various benchmark reasoning tasks. In addition, a novel benchmark knowledge graph with a *Clever Hans* bias is introduced to evaluate the alignment of link prediction methods with human oracle-regulated understanding of valid reasoning. The paper contributes by introducing preference-based human-in-the-loop learning to path-based link prediction in knowledge graphs. It aligns the model's behavior with human understanding of valid reasoning.

## 1 Introduction

Path-based link prediction methods learn paths within a knowledge graph that robustly reconstruct missing links between vertices (Das et al., 2018; Wan et al., 2020). The paths are the explanations for the link prediction, as they are a human-interpretable mapping of the input to the output (Bhowmik & de Melo, 2020; Bahr et al., 2025b; Wehner et al., 2023). For example, to predict that `Berlin` is in `Germany`, a path such as `Berlin - locatedInState` → `Brandenburg - partOfCountry` → `Germany` provides a clear, human-interpretable sequence of relations which maps the input entities to the predicted link. The paths are the observable reasoning of the link prediction (Lin et al., 2018). The user of a path-based link prediction method expect the reasoning to be valid (Copi, 1954) in the everyday sense, which we make precise below as semantically coherent. **Reasoning validity** in this paper is defined as oracle-aligned semantic relevance of a reasoning path. A path is valid when its steps are judged as semantically appropriate by the supervising oracle for explaining the target relation in the given domain. This notion of validity captures human-judged appropriateness of relational evidence and does not constitute logical entailment. We use the phrase *human understanding of valid reasoning* throughout to denote this oracle-judged semantic relevance. The qualifier *human understanding of* marks that validity here is a matter of human-judged appropriateness, not logical entailment. Where we write *valid reasoning* without the qualifier, the same operational sense is intended. These normative target paths reflect oracle-appropriate explanatory structure and avoid semantically irrelevant shortcuts. **Invalid reasoning** paths contain steps that are irrelevant to the conclusion because of spurious relations (i.e. edges that are factually correct but causally or semantically irrelevant to the predic-

tion). Correctness and reasoning validity are complementary but distinct axes of this problem. A model may produce a correct answer via semantically inappropriate reasoning (invalid) or follow a semantically appropriate path yet predict an incorrect entity instance (valid). Our evaluations are designed to test the tradeoff between both validity and correctness.

Consider the following example (cf. Figure 1). A path-based link prediction method predicts `Hungary` to be in `Europe` because `Hungary` consumes pepper from `Europe`. The conclusion that `Hungary` is in `Europe` is true. Also, the reasoning step that `Hungary` consumes pepper from `Europe` is true. However, `Hungary` consuming pepper from `Europe` is semantically irrelevant to whether `Hungary` is part of `Europe` or not. The reasoning is invalid. A preferable reasoning pattern is that `Hungary` is in the region of `Central Europe`, and `Central Europe` is in `Europe`. Here, the conclusion is true, and the reasoning is valid as it is relevant to the conclusion.

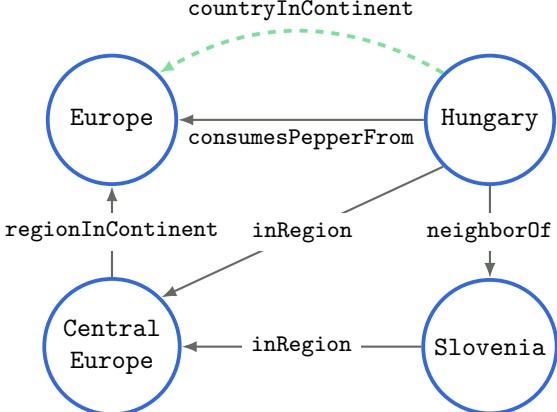

Figure 1: The figure depicts an excerpt from a knowledge graph about countries, regions, and continents. A path-based link prediction method predicts the missing link `countryInContinent` (green) from `Hungary` to `Europe` based on learned reasoning patterns connecting the two vertices.

Invalid reasoning patterns are likely to be learned by path-based link prediction methods if the patterns are frequent (dominant) and thus yield high accuracy (robust) on the train and validation set (Marconato et al., 2023), as will be demonstrated in Section 4. This is an alignment problem rooted in the discrepancy between what users expect the model to learn and what it actually learns (Christian, 2020). The users of the link prediction method expect it to learn a reasoning that reflects their knowledge, and thus understanding, of the semantic concept underlying the missing link (e.g., what it means for a country to be in a continent). However, the semantic concept is not explicitly included in the supervision signal given to the link prediction method while training, so it is not directly enforced during learning. Instead, invalid reasoning patterns, semantically unrelated to the classification goal, are potentially being learned if they frequently reconstruct a missing link on the training and validation sets. Meanwhile, these invalid reasoning patterns do not hold outside the training and validation splits; models that rely on them fail to generalize to real-world data. This is the **Clever Hans bias** (Lapuschkin et al., 2019): like the horse that seemed to do arithmetic by reading its trainer's cues, a link predictor can appear accurate by exploiting irrelevant, dataset-specific correlations instead of performing genuine, valid reasoning.

This paper proposes *LiEr*[1] (**L**earning **i**nteractively from **E**xplanations to **r**eason). *LiEr* is a path-based link prediction method that learns iteratively and interactively to reason (cf. Figure 2). It is optimized to reflect the supervising oracle's elicited preferences within the knowledge graph domain. This is achieved by learning a reward function iteratively from preference-based human feedback. A policy network proposes reasoning paths that maximize the expected reward.

---

[1]It is commonly believed that Li Er is the birth name of the semi-legendary philosopher Laozi, the founder of Taoism (Seidel & von Falkenhausen, 2008).

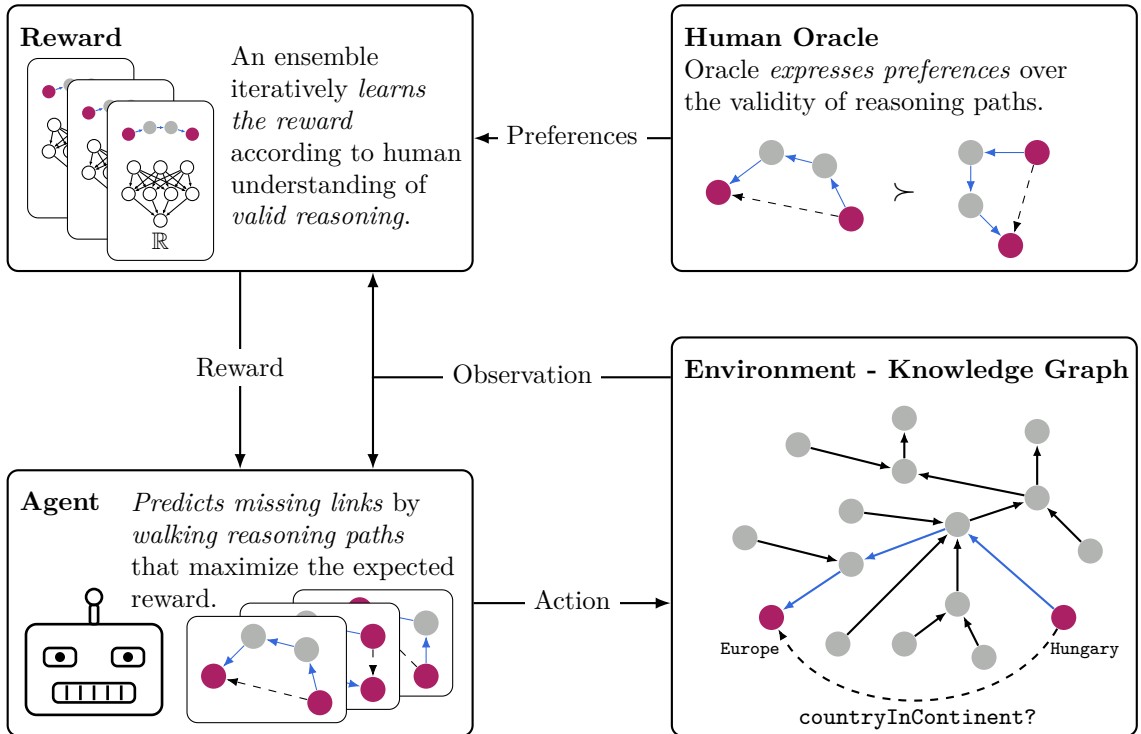

Figure 2: The schematic overview of *LiEr*. *LiEr* is a reinforcement learning agent that walks through a knowledge graph starting at a given head entity, finding the most plausible tail entity for a missing relation. The walk resembles a reasoning process aligned with human knowledge of the semantic concept underlying the missing link. *LiEr* realizes this alignment with the help of an ensemble of reward estimators trained on preferences from a single human oracle over the reasoning paths.

First, the paper introduces the formal notion of knowledge graphs and of the link prediction task. Next, it describes how the link prediction task is reformulated to a *Markov Decision Process (MDP)* (Bellman, 1957). The paper presents a non-stationary history-dependent policy to solve the *MDP*. It follows up by introducing the formal properties of the reward function, which approximate the oracle's understanding of valid reasoning. It describes how preference-based feedback over reasoning-pairs is collected from a human and how it is used to optimize the reward function towards rewarding valid reasoning. In addition, we argue why preference-based feedback is particularly suitable to approximate human oracle's of valid reasoning in path-based link prediction. Furthermore, the paper describes the training of the policy network. *LiEr* is evaluated on several knowledge graphs from reasoning-heavy domains to demonstrate that its predictive performance is competitive with state-of-the-art link prediction methods and achieving improved reasoning alignment. To assess this and ensure reproducible evaluation, we simulate the human oracle with an understanding of correctness and relation-level regions of interest (*ROI*s). Additionally, we validate these simulated oracle against a real annotator. These *ROI*s are an evaluation device and not a component of *LiEr*, which consumes only the resulting pairwise preferences and never the *ROI*s themselves (cf. Section 4.1). In addition, the paper introduces a novel benchmark knowledge graph, called *Clever Hans Countries*. *Clever Hans Countries* is designed to illustrate that *LiEr* aligns its reasoning with oracle's understanding of valid reasoning, leading to a performance boost in knowledge graphs with a high potential for link prediction models to learn *Clever Hans* biases (Lapuschkin et al., 2019). Finally, the paper provides observations and practical implications from preference-based feedback collection. The paper contributes by:

- Introducing preference-based human-in-the-loop learning, building on preference-based reinforcement learning (Christiano et al., 2017), to path-based link prediction in knowledge graphs.

- Demonstrating how to align the reasoning of a link prediction with oracle's understanding of valid reasoning.
- Describing the *Clever Hans* bias in knowledge graph completion and proposing a benchmark knowledge graph that captures the bias in a controlled manner.

## 2 Related Work

Research on path-based link prediction and human-in-the-loop learning inspired *LiEr*. Link prediction is organized in embeddings-based methods, symbolic logic-based methods, and neuro-symbolic link prediction methods (Zhang et al., 2021). Embedding-based methods, such as *TransE* (Bordes et al., 2013), *DistMult* (Yang et al., 2015), *ConvE* (Dettmers et al., 2018), and *RoEMF* (Lu et al., 2025) demonstrate good performance while scaling to large knowledge graphs. However, by embedding the knowledge graph into sub-symbolic space, embedding-based methods lose the expressive topology of the knowledge graph, resulting in uninterpretable predictions that do not necessarily adhere to principles of reasoning (Schramm et al., 2023; Schramm & Schmid, 2023; Wehner et al., 2026).

Symbolic reasoning and logic-based link prediction methods such as *AMIE+* (Galárraga et al., 2015), *ScaLeKB* (Chen et al., 2016), *SAFRAN* (Ott et al., 2021), and *AnyBURL* (Meilicke et al., 2024) learn explicit rules to predict missing links via reasoning. They operate entirely in the symbolic space, using the expressive topology of a knowledge graph. However, symbolic reasoning methods fall short if nuanced link prediction behavior for individual vertices is required, as they are limited in capturing special features of individual vertices (Schramm et al., 2023).

Neuro-symbolic link prediction methods, such as *NTP-λ* (Rocktäschel & Riedel, 2017), *NeuralLP* (Yang et al., 2017), *DRUM* (Sadeghian et al., 2019), and *NCRL* (Cheng et al., 2023) integrate symbolic and sub-symbolic methods to leverage the interpretability and structure of explicit rules while also benefiting from the scalability and statistical insights of sub-symbolic approaches. One particular line of research is path-based methods, such as *MINERVA* (Das et al., 2018), *Multi-Hop* (Lin et al., 2018), and *CURL* (Zhang et al., 2022). They reformulate the knowledge graph as a *MDP* (Bellman, 1957) to learn reasoning paths (Wan et al., 2020) that reconstruct plausible missing links, optimized through reinforcement learning.

Embedding-based, symbolic, and neuro-symbolic link prediction methods learn to predict missing links from patterns in the knowledge graph, optimizing for performance on the validation set, which serves as their only ground truth (Marconato et al., 2023). Thus, link prediction methods are vulnerable to learning invalid reasoning that does not generalize well to the real-world distribution (cf. Section 4.4), if invalid reasoning patterns consistently perform well on the training and validation set (Marconato et al., 2023). This is also called the *Clever Hans* bias (Lapuschkin et al., 2019) and can be understood as an alignment problem. In particular, there is a misalignment in the model's high performance on standard evaluation metrics while not learning valid reasoning patterns, which is what users actually expect from the model.

Artificial intelligence (*AI*) alignment research roots human intended goals and preferences into *AI* systems (Wiener, 1960; Christian, 2020; Gabriel, 2020; Teso et al., 2023; Ilievski et al., 2025). Developers and users of link prediction methods expect valid reasoning. However, the validity of reasoning depends on human knowledge of the semantic concepts underlying the missing link. The semantic concept is not explicitly part of the knowledge graph and, thus, can only be approximated. The approximation is realized by reconstructing tail mappings. The consequence is a misalignment between the *AI* system's intended goal, valid reasoning, and its realization via tail mapping reconstruction. This leads to unintended behavior of the link prediction method if the knowledge graph is exploitable with a *Clever Hans* bias (Lapuschkin et al., 2019). A large corpus of literature on incorporating human intent in machine-learning methods exists (Mosqueira-Rey et al., 2023; Najar & Chetouani, 2021). In particular, human-in-the-loop reinforcement learning approaches like *TAMER+RL* (Knox & Stone, 2011), deep reinforcement learning from human preferences (Christiano et al., 2017), *EXPAND* (Guan et al., 2020), and *MAPLE* (Mahmud et al., 2024) make significant progress in aligning the behavior of reinforcement agents in physical domains with human intent. Preference-based human-in-the-loop learning also showed successful applications in *natural language processing* with *Instruct-*

*GPT* (Ouyang et al., 2022). This paper aims to use preference-based human-in-the-loop learning to align the reasoning of a link prediction with human understanding of valid reasoning.

## 3 Aligning Link Prediction with Human Understanding of Valid Reasoning

In this paper, we introduce *LiEr*, a path-based link prediction method that is trained to align its internal reasoning with human understanding of valid reasoning. Section 3.1 formalizes the knowledge graph and defines the missing link prediction task of *LiEr*. This sets the foundation for the detailed description of *LiEr* in Section 3.2.1. In Section 3.2.1, the knowledge graph is translated into a *MDP*. Section 3.2.2 explains the policy learning method used to solve the *MDP*, given a reward function ensuring alignment with human reasoning (cf. Section 3.2.3). This section aims to fully describe the method behind *LiEr*'s preference-based reward alignment.

### 3.1 Defining the Knowledge Graph and Link Prediction Task for LiEr

**The setting of *LiEr* is a knowledge graph.** The knowledge graph $\mathcal{G}$ is defined as a directed, labeled, multigraph (Hogan et al., 2022; Das et al., 2018). Thus, the knowledge graph is a typed quiver (Assem et al., 2006)

$$\mathcal{G} = (V, E, \Sigma_V, \Sigma_E, h, t, \ell_V, \ell_E), \tag{1}$$

with a set $V$ of vertices $v \in V$ (i.e., entities/nodes), a set $E$ of edges $e \in E$ (i.e., relations/links), an alphabet $\Sigma_V$ with symbols for the vertices $\sigma_v \in \Sigma_V$, an alphabet $\Sigma_E$ for the edge symbols $\sigma_e \in \Sigma_E$. The head mapping $h : E \to V$ maps an edge $e$ to its head vertex $v_h$, and the tail mapping $t : E \to V$ maps an edge $e$ to its tail vertex $v_t$. The tail and head mapping enforce the directionality of the graph. The vertex language mapping $\ell_V : V \to \Sigma_V$ maps a vertex $v$ to its corresponding symbol $\sigma_v$, and the edge language mapping $\ell_E : E \to \Sigma_E$ maps an edge $e$ to its corresponding symbol $\sigma_e$ (Gallian, 2000; Harary et al., 1965).

To illustrate this thorough definition, consider a simple example knowledge graph with three vertices $V = \{v_1, v_2, v_3\}$. To assign meaningful names to these vertices, we define the vertex alphabet $\Sigma_V = \{\texttt{Poland}, \texttt{Central Europe}, \texttt{Europe}\}$. The vertex language mapping $\ell_V$ associates each vertex with its corresponding name. For example, $\ell_V(v_1)$ returns `Poland`.

Next, we define the edges that represent relations between vertices $E = \{e_1, e_2\}$. Their labels are provided by the edge alphabet $\Sigma_E = \{\texttt{neighbour}, \texttt{locatedIn}\}$. The edge language mapping $\ell_E$ associates each edge with its corresponding label. For instance, $\ell_E(e_1)$ returns `locatedIn`.

Finally, the connectivity between vertices is established by the head and tail mappings. The head mapping $h$ returns the starting vertex of an edge, while the tail mapping $t$ returns its ending vertex. Suppose that $h(e_1) = v_1$ and $t(e_1) = v_2$. With the help of this definition, we establish a structured way to retrieve the fact that `Poland is located in Central Europe`. Additionally, this structured approach enables the subsequent translation of the knowledge graph into a traversable *MDP*.

**The goal of LiEr is to predict missing links.** *LiEr* learns a mapping

$$p : V \times \Sigma_E \to V, \tag{2}$$

that predicts for any given query $q$, consisting of a head vertex $v_h^q$ and edge symbol $\sigma_e^q$, the most plausible tail vertex $v_t^q$, referred to as the *answer*. The query $q$ is frequently represented as a triple $(v_h^q, \sigma_e^q, ?)$. The question mark in $q$ denotes the return value of $p$, which is the most plausible tail vertex.

The link prediction mapping $p$ is selected from the set of all possible mappings $p \in P$ to approximate the tail mapping $t$ as accurately as possible on a validation edge set $E_{valid} \subset E$. Satisfying Equation 3 objectivizes the link prediction mapping $p$ towards a consistently high performance over unseen queries (i.e., validation data). Thus, the link prediction mapping $p$ aims to learn reasoning patterns that generalize well to all possible and correct query-answer pairs

$$\arg\max_{p \in P}(|\{e | \forall e \in E_{valid} : t(e) = p(h(e), \ell_E(e))\}|). \tag{3}$$

The advantage of the tail link prediction mapping $p$ over tail mapping $t$ is that $p$ does not require an edge between $v_h^q$ and $v_t^q$ to determine $v_t^q$ as the correct tail vertex. The link prediction mapping $p$ solely relies on a head vertex $v_h^q$ and an edge symbol $\sigma_e^q$. This allows $p$ to return $v_t^q$ even if an edge $e$ with the symbol $\sigma_e^q$ between $v_h^q$ and $v_t^q$ is missing.

Reconsider the simple knowledge graph from the previous example. Suppose that we want to determine the continent in which `Poland` is located. In the current graph, the corresponding edge (i.e., the relation `locatedIn`) linking `Poland` to its continent is missing. Thus, the tail mapping cannot be used to retrieve an answer.

To address this gap, we introduce the link prediction mapping $p$. The mapping takes a query in the following manner: $q = (\texttt{Poland}, \texttt{locatedIn}, ?)$, as input. The link prediction mapping $p$ processes the query $q$ based on its learned patterns and returns the most plausible vertex candidate for the missing link, ideally returning `Europe` as the answer.

As described in the introduction, this paper proposes an approach that allows for human-in-the-loop learning of the link prediction mapping $p$. The aim is to iteratively improve the reasoning employed by $p$ to arrive at an answer $v_t^q$ with the help of human feedback.

### 3.2 Predicting Missing Links with LiEr by Learning Valid Reasoning from Human Feedback

The following formulates the link prediction task as an *MDP* (Bellman, 1957), similar to previous research on path-based link prediction methods (Das et al., 2018; Lin et al., 2018; Wan et al., 2020). A knowledge graph can easily be understood as an MDP, by looking at its nodes as states and its edges as actions. This enables the optimization of the link prediction mapping $p$ according to Equation 3 via reinforcement learning. For that reason, Section 3.2.1 describes how the knowledge graph is transformed into an *MDP*. Section 3.2.2 details the policy learning strategy used to solve the *MDP*, which leverages a reward function specifically designed to align with human reasoning, explained in Section 3.2.3. Finally, the training procedure for *LiEr* is outlined, describing the optimization of its preference-based reward.

### 3.2.1 From Knowledge Graph to Markov Decision Process

The *Markov Decision Process* is defined as a 4-tuple $MDP = (\mathcal{S}, \mathcal{A}, \delta, \mathcal{R})$.

The *MDP* holds a state space $\mathcal{S} \subseteq \{V \times V \times \Sigma_E\}$. A state $s_c \in \mathcal{S}$ at step $c$ is defined as $s_c = (v_c, v_h^q, \sigma_e^q)$. It consists of the vertex $v_c$ and $(v_h^q, \sigma_e^q)$, which is the input to $p$ (cf. Equation 2).

A set of actions $\mathcal{A}_{c_s} \subseteq \mathcal{A}$ is available at each state $s_c$.

$$\mathcal{A}_{c_s} = \{(v_c, e, v) | \forall e \in E, \forall v \in V : h(e) = v_c \wedge t(e) = v\} \cup \{(v_c, e_{loop}, v_c)\} \tag{4}$$

The 3-tuple $(v_c, e, v)$ is called an action $a$. Intuitively, an agent can *walk* any of the outgoing edges $e$ from vertex $v_c$ to arrive at $e$'s tail node $v$. In addition, a loop action $(v_c, e_{loop}, v_c)$ is available. The loop action allows staying at a vertex $v_c$. This is useful in cases where staying at $v_c$ maximizes the expected reward.

The mapping $\delta : \mathcal{S} \times \mathcal{A} \to \mathcal{S}$ defines the transition from the current state $s_c$ to the follow-up state $s_{c+1}$, given action $a_c = (v_c, e, v_{c+1})$:

$$\delta(s_c, a_c) = (v_{c+1}, v_h^q, \sigma_e^q) = s_{c+1}. \tag{5}$$

The tail vertex $v_{c+1}$ of the selected action $a_c$ becomes the current vertex of the following step $c+1$. The transition probability $P(s_{c+1}|s_c, a_c) = 1$ is fully deterministic.

Finally, the *MDP* includes a reward function $\mathcal{R} : \mathcal{S} \times \mathcal{A} \to \mathbb{R}$. The reward $\mathcal{R}$ takes as input a state $s \in \mathcal{S}$, and an action $a \in \mathcal{A}$ and returns a numeric reward. Section 3.2.3 describes how $\mathcal{R}$ is parameterized and trained to be aligned with human understanding of valid reasoning.

Once the knowledge graph is transformed into an *MDP*, a policy has to be defined that describes and governs the walk through the *MDP* to optimize the link prediction mapping defined in Equation 3 that returns the most plausible answers to a link prediction query. This process is called solving the *MDP*.

The reframing of the link prediction task as a *MDP* is now fully described. The following section introduces how the *MDP* is solved to predict missing links.

### 3.2.2 Solving the Markov Decision Process

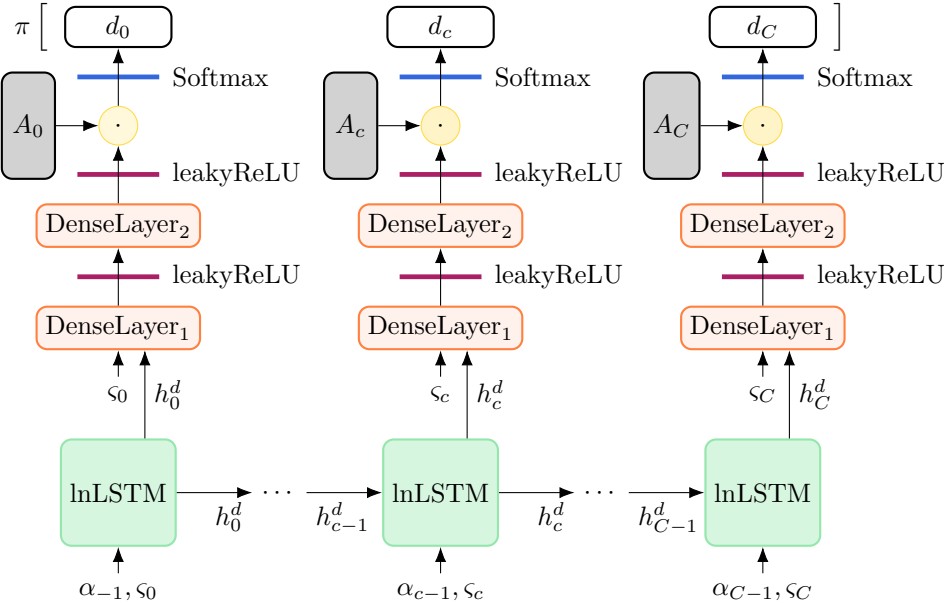

Figure 3: The policy network architecture of *LiEr* operates as follows. At each step $c$, with $0 \leq c \leq C$, the probability distribution $d_c$, as part of the policy $\pi$, is calculated using a layer-normalized long-short term memory ($lnLSTM$). This is followed by two dense layers that utilize leaky ReLU activations. The output from these layers is then multiplied ($\cdot$) by the stacked embeddings of all available actions $A_c$. Finally, this results in the probability distribution $d_c$ through a softmax function (cf. Glossary 16).

The following describes the solving process of the *MDP*. To get an explicit solution, a non-stationary history-dependent policy $\pi = (d_1, \ldots, d_c)$ is implemented. To solve the policy $\pi$, at each step $c$, a policy network calculates a probability distribution $d_c$ over all available actions $\mathcal{A}_{c_s}$ defined by:

$$h_c^d = \text{lnLSTM}(h_{c-1}^d, [\alpha_{c-1}; \varsigma_c]) \tag{6}$$

$$d_c = \text{Softmax}(A_c \cdot \text{leakyReLU}(W_2 \cdot \text{leakyReLU}(W_1 \cdot [h_c^d; \varsigma_c] + B_1) + B_2)). \tag{7}$$

The recurrent neural network architecture in Equation 6 and 7 describes the policy network. The recurrent architecture enables the policy network to consider previously encountered states and actions while choosing the next action (cf. Figure 3). At step $c$, a layer normalized long short-term memory ($lnLSTM$) (Ba et al., 2016) calculates the history $h_c^d \in \mathbb{R}^m$ (cf. Equation 6). The $lnLSTM$ takes two inputs. The first input is the previous history $h_{c-1}^d$. The second input is a stacking of the previously executed actions' ($a_{c-1}$) embedding $\alpha_{c-1} \in \mathbb{R}^n$ and the current states' ($s_c$) embedding $\varsigma_c \in \mathbb{R}^l$. The numbers $m, n, l \in \mathbb{N}$ are the lengths of the embeddings. Next, the policy network stacks $h_c^d$ with $\varsigma_c$. The stacked embedding is first passed to a dense layer (Goodfellow et al., 2016) with weights $W$, biases $B$, and a leaky rectified linear ($leakyReLU$) activation (Redmon et al., 2016). This is followed by another dense layer that also uses a $leakyReLU$ activation. The embeddings of all available actions $\mathcal{A}_{c_s}$ are stacked in $A_c \in \mathbb{R}^{|\mathcal{A}_{c_s}| \times n}$ and multiplied with the output of the previous dense layer. This results via a *softmax* function (Goodfellow et al., 2016) in a probability distribution $d_c \in \mathbb{R}^{|\mathcal{A}_{c_s}|}$ over all available actions $\mathcal{A}_{c_s}$ (cf. Equation 7).

To calculate the policies for every step, the agent starts at the head vertex of the query $v_h^q$ and chooses an action $a_c \in \mathcal{A}_{c_s}$ according to the probability distribution $d_c$. In cases of exploration, this means that an action is sampled from all available actions according to the probability distribution. This is the dominant strategy while training. In exploitation cases, such as the evaluation, the action with the highest probability

is chosen. This is the fuzzy reasoning step of *LiEr*. The transition is applied as described in Equation 5, and the process is repeated at the next state with its available actions and an updated history. The agent stops after a finite and predefined number of repetitions $C$ (i.e., steps), with $0 \leq c < C$. The final vertex in the final state $v_C \in s_C$ after all steps are executed is the return vertex $v_t^q$ of the link prediction mapping $p$ (cf. Equation 3).

Let the composition of all reasoning steps $\kappa_c = (s_c, a_{c-1})$ be a path $w = (\kappa_1, ..., \kappa_c, ..., \kappa_C)$. The path reflects the reasoning applied by the link prediction mapping $p$ to arrive at the answer. It maps the query to the answer. Thus, the path is the policy applied to solve the *MDP* and the explanation of the link prediction.

**On aligning the reasoning steps with human understanding of valid reasoning.** Notably, $A_c$ binds the sub-symbolic policy network to the topology of the knowledge graph, a syntactically correct and symbolic system. This results in individual steps of the agent being true and interpretable. Meanwhile, the composition of all steps may still lead to a wrong query answer. In particular, the composition of all steps may not reflect human understanding of valid reasoning regarding their knowledge about the semantic concept underlying the missing link.

The policy network's parameters are trained to maximize the expected reward. We use this to align *LiEr*s' reasoning paths with human understanding of valid reasoning.

The next section explains how the reward $\mathcal{R}$ is modeled to guide the policy network toward reasoning that matches human understanding of the query's underlying concept.

### 3.2.3 Aligning the Policy Network with Human Understanding of Valid Reasoning via the Reward

This section proposes the alignment of the policy network with valid reasoning, by the reward $\mathcal{R}$. Domain experts generally know valid reasoning within their domain. However, they may not know how domain knowledge is formalized within the knowledge graph. For example, the domain expert may know that `Hungary` is in `Europe` because `Hungary` is in `Central Europe` and `Central Europe` is in `Europe`. However, they may not know that "`Hungary` is in `Central Europe`" is formalized via the `locatedIn` relationship (i.e., `locatedIn(Hungary, Central Europe)`). This makes it difficult for domain experts to give rules or examples according to the formalization enforced by the knowledge graph. In addition, some domain knowledge is tacit. Valid reasoning patterns may be difficult for the expert to verbalize. However, if the domain experts see examples, they can judge the correctness and validity.

For that reason, we propose a preference-based feedback approach that automatically creates reasoning examples for the domain expert to judge. This leaves it to the domain expert to establish a ranking over the reasoning examples, going from valid and correct to incorrect examples. However, asking domain experts to rank many reasonings by categories or numeric values is cumbersome. Thus, the paper's approach uses pairwise preference collection, making it convenient for the expert to create a ranking. The preference-based approach enables a user-friendly collection of preferences and thus provides an efficient way to learn the human oracle's $\Lambda$ understanding of valid reasoning.

Central to the pairwise preference alignment approach is the reward $\mathcal{R}$. The rewards are computed using an ensemble of recurrent neural networks (Dietterich et al., 2002), each computing $r_c^i$ for the individual reasoning step $\kappa_c$. The computation is carried out as follows

$$h_c^r = \text{lnLSTM}\left(h_{c-1}^r, \varkappa_c\right), \tag{8}$$

$$r_c^i = \text{batchNorm}\left(\text{leakyReLU}\left(W_2 \cdot \text{leakyReLU}\left(W_1 \cdot [h_c^r; \varkappa_c] + B_1\right) + B_2\right)\right), \tag{9}$$

where $\varkappa \in \mathbb{R}^{l+n} = [\varsigma; \alpha]$ is the vector representation of $\kappa$, and $l$ is the length of the state embedding $\varsigma$, $n$ the length of the action embedding $\alpha$. The reward is normalized over the batch (*batchNorm*) to stabilize the training. The statistics are computed per batch. Because the two paths of a preference pair are scored within the same batch and have the same fixed length $C$, the normalization subtracts the same mean and divides by the same standard deviation for both summed rewards $\sum_{\forall \varkappa \in \omega} \hat{r}(\varkappa)$. Therefore, batch-normalization does not change which path has the larger summed reward. The only effect of batch-normalization is that it

changes the numerical scale of the logits that the Bradley-Terry model compares. The Bradley-Terry model operates on these rescaled logits to compute preference probabilities between the two paths (cf. Equation 13).

The training requires aligning the reward with the help of labels annotated by a human oracle. To make the alignment process as efficient as possible, we select samples of paths with the highest uncertainty in the reward for feedback. This ensures a large impact on the reward training. To calculate the uncertainty in the reward, a reward ensemble is used (Dietterich et al., 2002). The final reward $\hat{r}_c$ is the mean over the rewards $r_c^i$ of each ensemble member $i$.

The reward ensemble is trained to enforce properties that approximate valid reasoning, based on a human's knowledge of the semantic concept underlying a missing link. The following paragraphs formally introduce those properties necessary for aligning *LiEr* with human understanding of valid reasoning.

**Properties of $\hat{r}$.** The reward $\hat{r}$ is optimized to incentivize choosing paths based on the preference provided by a human oracle $\Lambda$, following the approach outlined by (Christiano et al., 2017).

Let the reasoning paths $w_{\sigma_e}^1 = (\kappa_1^1, ..., \kappa_c^1, ..., \kappa_C^1)$ and $w_{\sigma_e}^2 = (\kappa_1^2, ..., \kappa_c^2, ..., \kappa_C^2)$ result from a maximum of two different queries $q_1$ and $q_2$ with the same edge symbol $\sigma_e$. The embedded reasoning steps $\varkappa^1$ and $\varkappa^2$ of $w_{\sigma_e}^1$ and $w_{\sigma_e}^2$ are stored in $\omega_{\sigma_e}^1 = (\varkappa_1^1, ..., \varkappa_c^1, ..., \varkappa_C^1)$ and $\omega_{\sigma_e}^2 = (\varkappa_1^2, ..., \varkappa_c^2, ..., \varkappa_C^2)$.

Given two reasoning paths $w_{\sigma_e}^1$ and $w_{\sigma_e}^2$, $\hat{r}$ satisfies the following: The reasoning path $w_{\sigma_e}^1$ shall receive a higher reward $\hat{r}$ compared to $w_{\sigma_e}^2$, if the human oracle $\Lambda$ favors $w_{\sigma_e}^1$ over $w_{\sigma_e}^2$, as detailed in Equation 10. Conversely, the path $w_{\sigma_e}^2$ will receive a higher reward, if $\Lambda$ prefers $w_{\sigma_e}^2$ over $w_{\sigma_e}^1$, as indicated in Equation 11. If $\Lambda$ is indifferent between the two reasoning paths, $w_{\sigma_e}^1$ and $w_{\sigma_e}^2$ shall receive similar rewards, as stated in Equation 12.[2]

$$\sum_{\forall \varkappa^1 \in \omega_{\sigma_e}^1} \hat{r}(\varkappa^1) > \sum_{\forall \varkappa^2 \in w_{\sigma_e}^2} \hat{r}(\varkappa^2) \text{ if } w_{\sigma_e}^1 \succ_\Lambda w_{\sigma_e}^2 \tag{10}$$

$$\sum_{\forall \varkappa^1 \in \omega_{\sigma_e}^1} \hat{r}(\varkappa^1) < \sum_{\forall \varkappa^2 \in w_{\sigma_e}^2} \hat{r}(\varkappa^2) \text{ if } w_{\sigma_e}^1 \prec_\Lambda w_{\sigma_e}^2 \tag{11}$$

$$\sum_{\forall \varkappa^1 \in \omega_{\sigma_e}^1} \hat{r}(\varkappa^1) \approx \sum_{\forall \varkappa^2 \in w_{\sigma_e}^2} \hat{r}(\varkappa^2) \text{ if } w_{\sigma_e}^1 \sim_\Lambda w_{\sigma_e}^2 \tag{12}$$

Note that the preferences constrain only the *sum* of the per-step rewards along a path; a single path-level preference is consistent with many per-step decompositions. The learned per-step rewards therefore act as a policy-shaping signal, and we assert validity at the level of the whole reasoning path rather than claiming that an individual step reward certifies the validity of that step.

**Properties of $\Lambda$.** Two major factors lead the human oracle $\Lambda$ while expressing preferences over paths:

1. *Correctness*: The answer is correct.

2. *Validity*: The path of reasoning steps is judged semantically relevant and appropriate (cf. Section 1)

Let us discuss the properties of $\Lambda$ given the example from Table 1. In *Ex.1*, the reasoning is valid: Hungary is in Central Europe, and Central Europe is in Europe, so the conclusion that Hungary is in Europe is correct. In *Ex.2*, the conclusion is also correct, but the reasoning is invalid because it relies on Hungary consuming pepper from Europe, which is unrelated to the claim. Meanwhile, *Ex.3* concludes that Hungary is in Austria, which is incorrect. The human oracle prefers the correct and valid reasoning *Ex.1* over the correct but invalid reasoning *Ex.2* and reasoning *Ex.2* over the incorrect and invalid reasoning *Ex.3*. This results in the preference order $Ex.1 \succ Ex.2 \succ Ex.3$.

---

[2]The preferences constrain the sum of the per-step rewards along a path, and a single path-level preference is consistent with many per-step decompositions. The learned per-step rewards therefore act as a policy-shaping signal. We assert validity at the level of the whole reasoning path, not of individual steps.

---

(*Ex.1*)  `inContinent(Hungary, Europe) ← inRegion(Hungary, Central Europe) ∧ inContinent(Central Europe, Europe)`

---

(*Ex.2*)  `inContinent(Hungary, Europe) ← consumesPepperFrom(Hungary, Europe)`

---

(*Ex.3*)  `inContinent(Hungary, Austria) ← hasNeighbor(Hungary, Austria)`

---

Table 1: The three exemplary reasonings *Ex.1*, *Ex.2*, and *Ex.3* are possible results for the query (`Hungary, inContinent, ?`). The correct answer to the query is `Europe`. *Ex.1* and *Ex.2* arrive at this answer. However, *Ex.1*'s reasoning is valid and *Ex.2*'s is not.

The reward ensemble $\hat{r}$ is optimized to approximate the preference ordering of $\Lambda$ by rewarding preferred reasoning higher compared to less preferred ones. Thus, $\hat{r}$ learns to reward reasoning steps by the *correctness* and *validity* of the resulting path. Crucially, the reward depends only on these two criteria as judged by $\Lambda$. It contains no term that rewards shorter paths or penalizes longer ones, and path length affects the reward only insofar as it bears on validity. Unlike correctness-only path-walkers, for which shorter trajectories tend to be discovered earlier and accumulate reward sooner, *LiEr* therefore has no intrinsic bias toward short paths and can assign higher reward to a longer valid path than to a shorter invalid one (cf. Sections 4.3 and 4.4).

**Preference collection.** The preference collection from a human oracle $\Lambda$ requires pairs of reasoning paths. For that reason, pairs of paths $(w^1_{\sigma_e}, w^2_{\sigma_e})$ are sampled at each training epoch. The pairs are sampled such that the feedback required from $\Lambda$ is minimized. This is done by selecting the $n \in \mathbb{N}$ pairs with maximum variance in the reward across all ensemble members (Christiano et al., 2017). Intuitively, pairs are selected and shown to $\Lambda$ for feedback, for which the reward ensemble is most uncertain on how to reward them.[3] This is done in the hopes of eradicating uncertainties and closing corresponding blind spots in the reward function. Finally, they are presented to the human oracle $\Lambda$ (cf. Figure 4). $\Lambda$ expresses preferences ($\succ \veebar \prec \veebar \sim$) over the pairs in alignment with their understanding of *correctness* and *validity*. Every pair $(w^1_{\sigma_e}, w^2_{\sigma_e})$ and preference ($\succ_\Lambda \veebar \prec_\Lambda \veebar \sim_\Lambda$) is stored in a database $D$ for training the reward ensemble. In the main benchmark experiments, the human oracle $\Lambda$ is simulated with relation-level *ROI*s for reproducibility. Section 4.5 reports runs annotated by a real human (cf. Section 4.1). *LiEr* consumes only the preferences expressed by $\Lambda$ and never the *ROI*s, so the *ROI*s are not part of the method.

**Fitting the Reward ensemble.** At the core of the reward ensemble training is the loss function (cf. Equation 14). The loss function is similar to the loss proposed by (Christiano et al., 2017). It is based on the *Bradley-Terry* model (Bradley & Terry, 1952). The *Bradley-Terry* model estimates a score function by observing pairwise preferences.

$$\hat{P}[w^1 \succ w^2] = \frac{\exp\left(\sum_{\forall \varkappa^1 \in \omega^1} \hat{r}(\varkappa^1)\right)}{\exp\left(\sum_{\forall \varkappa^1 \in \omega^1} \hat{r}(\varkappa^1)\right) + \exp\left(\sum_{\forall \varkappa^2 \in \omega^2} \hat{r}(\varkappa^2)\right)} \tag{13}$$

$$loss(\hat{r}) = - \sum_{\forall (w^1_{\sigma_e}, w^2_{\sigma_e}) \in D} (\mu * \log(\hat{P}[w^1_{\sigma_e} \succ w^2_{\sigma_e}]) + (1 - \mu) * \log(\hat{P}[w^2_{\sigma_e} \succ w^1_{\sigma_e}])) \tag{14}$$

$$\text{, with } \mu = \begin{cases} 1.0 \text{ if } w^1_{\sigma_e} \succ_\Lambda w^2_{\sigma_e} \\ 0.0 \text{ if } w^1_{\sigma_e} \prec_\Lambda w^2_{\sigma_e} \\ 0.5 \text{ if } w^1_{\sigma_e} \sim_\Lambda w^2_{\sigma_e} \end{cases} \tag{15}$$

First, the probability of $\hat{r}$ preferring path $w^1$ over path $w^2$ and the probability of $\hat{r}$ preferring path $w^2$ over $w^1$ are computed (cf. Equation 13). Next, each probability is multiplied by the corresponding weighting

---

[3] A single reward model would provide no disagreement signal and hence no uncertainty estimate, so the ensemble is required by this selection step and not an optional component. Additionally, averaging over the members reduces the variance of the reward estimate.

factor. $\mu$ is used for the probability related to $w^1$, while $(1 - \mu)$ is used for the probability related to $w^2$ (cf. Equation 14). The weighting factor $\mu$ is defined such that it equals 1 when $\Lambda$ prefers $w^1$ over $w^2$, and it equals 0 when $w^2$ is preferred over $w^1$ (cf. Equation 15). If $\Lambda$ is indifferent between the two paths, then $\mu$ is set to 0.5. This results in a loss that optimizes $\hat{r}$ to return a high reward if a path is preferred by $\Lambda$ and a low reward if a path is not preferred by $\Lambda$. Thus, $\hat{r}$ is optimized to approximate $\Lambda$'s preference ordering over all possible paths. The loss is calculated for all pairs in the database $D$ and back-propagated through $\hat{r}$.

It is fully described how *LiEr* predicts links between two vertices via valid reasoning paths. The following section presents the experimental results.

## 4  Evaluation

The evaluation of *LiEr* is structured to test its ability to learn valid reasoning patterns from human feedback and to compare its predictive performance with established link prediction methods. To that end, *LiEr* is first benchmarked on the *Countries* knowledge graphs (Bouchard et al., 2015; Rocktäschel & Riedel, 2017) and its tasks. These are controlled environments, designed to test the reasoning capabilities of link prediction methods in varying degrees of difficulty. Similar to *MNIST* (Deng, 2012) in image classification, the *Countries* knowledge graph is narrowly scoped, highly structured, and easily understandable by humans. This makes it ideal for testing whether a method captures the basic mechanism it is designed for. In this case, we want to test *LiEr* on multi-step reasoning in knowledge graphs, without the confounding factors of scale, noise, or ambiguity. However, like *MNIST* or the *Cats vs. Dogs* (Parkhi et al., 2012) datasets, *Countries* is, due to its limited scope, not sufficient to assess real-world feasibility.

To assess performance in more complex and realistic settings, we extend the evaluation to three additional knowledge graphs: *family* (Yang et al., 2017), and the *NELL* (Mitchell et al., 2018) subsets *locations*, and *sports*. These datasets provide more heterogeneous structures and less constrained reasoning patterns, and serve to evaluate *LiEr*´s ability to generalize beyond synthetic benchmarks.

Finally, we are going to evaluate *LiEr*´s key claim: that preference-based feedback enables it to learn reasoning patterns aligned with the oracle's understanding of valid reasoning, even in the presence of spurious relations. To that end, the paper introduces the *Clever Hans Countries* knowledge graph, which contains intentionally constructed spurious correlations that are predictive on the training and validation data but fail to generalize to the test distribution.

In addition to predictive performance, we analyze the dynamics of *LiEr*´s interactive learning procedure. This includes measuring how much human feedback is required to train a well-performing model and how preference-based training behaves in the presence of label noise or inconsistent feedback.

### 4.1  Evaluation Setup

**Implementation Details.**  *LiEr* is implemented in *PyTorch 1.10*. All evaluations were run on a workstation with an *AMD Ryzen Threadripper 2920X*, two *GeForce RTX 2080 TI*, and 64GB RAM. The policy network (cf. Equations 6 and 7)) is trained using *REINFORCE* (Williams, 1992). The same adaptations as in Das et al. (2018) are used to calculate the expected reward and cost function. An additive control variate baseline reduces variance in the expected reward (Das et al., 2018; Evans et al., 2000; Hammersley, 2013). Furthermore, an entropy regularization term is added to the cost function to encourage a diverse sampling of reasoning paths during training time (Das et al., 2018). In addition, dropout is applied to all layer types and the available actions $\mathcal{A}_c^s$ to reduce overfitting (Goodfellow et al., 2016) and, again, to encourage a diverse sampling of reasoning paths during training time. The reward ensemble (cf. Equation 9) is trained with the *ADAM* optimizer (Kingma & Ba, 2015) and the Bradley-Terry loss (cf. Equation 14), an early stopping mechanism, and dropout is applied at every layer type to reduce overfitting (Goodfellow et al., 2016) [4]. During policy-network training, the reward ensemble is evaluated in inference mode. Its batch-normalization layers use their frozen running statistics and not per-batch statistics, and dropout is

---

[4] A hyperparameter optimization study was conducted using *Optuna* (Akiba et al., 2019) for all hyperparameters, employing the *TPESampler* (Bergstra et al., 2011) to determine the best configurations for each dataset. For hyperparameter configurations specific to each knowledge graph, please refer to LiEr´s Github repository.

Table 2: Per-step training cost on two *RTX 2080 Ti*, averaged over five runs. *MINERVA* performs no reward-model fitting (n/a). *LiEr*'s policy update is architecturally identical to *MINERVA*'s, so the per-epoch cost is shared; the per-refit cost is the measured training-time difference, attributed to reward-model fitting, and its frequency is a hyperparameter. Inference latency is comparable for both methods, as the reward ensemble is not invoked at inference.

| Method | Clever Hans | | Sports | |
| | Policy / epoch (s) | Reward / refit (s) | Policy / epoch (s) | Reward / refit (s) |
| --- | --- | --- | --- | --- |
| *MINERVA* | 13.4 | – | 11.5 | – |
| *LiEr* | 13.6 | 7.5 | 11.6 | 2.0 |

disabled. The reward $\hat{r}$ assigned to a trajectory is therefore deterministic and independent of the batch it is evaluated in, so it introduces no additional non-stationarity into the *REINFORCE* gradient estimate.

**Computational Cost.** We compare *LiEr* against *MINERVA*, the most directly comparable path-based baseline, to quantify the overhead of preference-based training. Table 2 reports the breakdown. *LiEr*'s per-epoch policy update is on par with *MINERVA*. The additional cost is periodic reward-model refitting, the frequency of which is a hyperparameter. Each refit adds roughly 2 to 8 seconds on our hardware, the higher figure reflecting the larger preference set accumulated on *Clever Hans Countries*. The total training overhead is therefore the number of refits multiplied by the per-refit cost, a modest one-time cost relative to the alignment and robustness gains reported below. At inference, both methods are path-walking policies of comparable architecture and the reward ensemble is not used, so per-query latency is comparable.

**Simulating the Oracle with Regions of Interest.** We simulate the oracle's preferences using predefined regions of interest (*ROIs*) defined per relation type in each knowledge graph to scale the feedback process under limited human availability. *ROIs* in knowledge graphs serve a role analogous to *ROIs* in image processing (Brinkmann, 2008): they define structured subregions that are relevant to the prediction task at hand. In our case, each *ROI* represents an abstract, valid reasoning pattern that plausibly reconstructs the missing link between head and tail entities via multi-hop paths. These regions are derived from domain knowledge and reflect reasoning patterns that align with human understanding of validity [5].

For example, in the *Countries* knowledge graph, a typical *ROI* for the relation `locatedIn` is the reasoning pattern:

$$\text{Country} \rightarrow \texttt{locatedIn} \rightarrow \text{Region} \rightarrow \texttt{locatedIn} \rightarrow \text{Continent}$$

Another *ROI* looks as follows:

$$\text{Country} \rightarrow \texttt{neighborOf} \rightarrow \text{Country} \rightarrow \texttt{locatedIn} \rightarrow \text{Continent}$$

Each *ROI* defines a reasoning template considered valid for the corresponding relation.

We generate preferences by assuming the oracle prefers reasoning paths that (i) follow an *ROI* over those that do not, and (ii) lead to the correct tail entity over those that do not. These assumptions let us derive preference labels automatically, that reflect correct and valid paths. The *ROI* and correctness rule are an implicit aggregation policy that defines a single oracle's preference function. The generated labels do not encode multi-human conflict or heterogeneity.

The *ROIs* are not a component of *LiEr*. *LiEr* consumes only pairwise preferences over reasoning paths and never accesses the *ROIs* directly. We use *ROI*-derived preferences to instantiate the human oracle

---

[5]The construction of *ROIs* is non-trivial and only feasible for relation types with well-defined semantics in structured domains. As a result, the creation of *ROIs* constrained the choice of benchmark knowledge graphs and directly influenced the selection of the seven knowledge graphs used in this work.

so that our experiments are reproducible and do not require human annotators for every training run. Simulating the oracle with a known preference model is standard practice in preference-based reinforcement learning (Christiano et al., 2017; Lee et al., 2021); the assumption that the simulated oracle follows the *ROI*s is the analogue of the scripted-oracle assumption used there. As we show in Section 4.5 (Figure 5), *LiEr* does not require the simulated oracle to follow the *ROI*s perfectly. Performance degrades only gradually as the oracle's preferences deviate from the *ROI*s. This indicates that the method learns from the preference signal itself rather than from the *ROI* templates. All benchmark results reported in this paper (Tables 4 to 11 and Figure 5) are produced with *ROI*-derived preferences from the simulated oracle. The exception is the human-feedback validation in Section 4.5, where preferences are collected from a real annotator.

A more detailed description of how *ROI*s are defined for each knowledge graph is provided in the respective dataset sections.

**Comparison Models.** The predictive performance of *LiEr* is evaluated against several representative link prediction models that capture different methodological paradigms. As a neuro-symbolic baseline, we compare against *MINERVA* (Das et al., 2018), a non-interactive, path-based model that uses reinforcement learning to discover multi-hop reasoning paths in knowledge graphs. In addition, we include *Neural Theorem Provers (NTP)* and its improved variant *NTP-λ* (Rocktäschel & Riedel, 2017), which perform differentiable logical inference via end-to-end training. We also consider *Neural Logic Programming (NeuralLP)* (Yang et al., 2017), which combines logic programming with differentiable rule learning. All three models provide interpretable reasoning chains and represent widely evaluated neuro-symbolic baselines for multi-hop reasoning. Their official implementations were used[6], with default hyperparameters unless dataset-specific configurations were available.

To contextualize *LiEr*'s performance among embedding-based models, we further compare against *TransE* (Bordes et al., 2013), *DistMult* (Yang et al., 2015), and *ConvE* (Dettmers et al., 2018). These models learn low-dimensional vector representations of entities and relations and are widely used due to their scalability and empirical performance. They do not model explicit reasoning chains but serve as baselines for general link prediction tasks. All embedding-based models were trained using the *PyKEEN* (Ali et al., 2021) framework[7], which offers standardized training pipelines and reproducible implementations of knowledge graph embedding methods.

The selection of comparison models aims to span the spectrum from reasoning to sub-symbolic embedding-based approaches, thereby providing a meaningful baseline for assessing *LiEr*'s reasoning capabilities, alignment behavior, and overall performance.[8]

## 4.2 Evaluation Metrics

The following quantitative evaluation is based on two metrics: *Mean Reciprocal Rank (MRR)* and *Mean Reciprocal Localisation Rank (MRLR)*.

**MRR.** is a standard metric in link prediction tasks that measures the average inverse rank of the first correct answer (Fuhr, 2018; Ali et al., 2021). For each query $u$, the model generates a ranked list of candidate entities, and the position of the first correct entity is recorded. A high MRR score (closer to 1) indicates that correct answers are consistently ranked near the top, reflecting strong predictive performance. Formally, the

---

[6]MINERVA: `https://github.com/shehzaadzd/MINERVA`, NTP/NTP-λ: `https://github.com/uclnlp/ntp`, NeuralLP: `https://github.com/fanyangxyz/Neural-LP`

[7]PyKEEN: `https://github.com/pykeen/pykeen`

[8]We note that LiEr and the non-interactive baselines do not consume identical supervision. LiEr additionally receives pairwise validity preferences, whereas the baselines are trained only on the observed graph. This asymmetry is by design and is the defining property of the human-in-the-loop method. Our comparison asks whether consuming this preference signal yields a benefit, not whether the methods are matched in their inputs. A complementary control that equips a baseline with the same semantic signal in its most direct form, a fixed reward for ROI-template overlap, is reported in Section 4.4.1, and it isolates the contribution of the preference-learning mechanism from that of the signal itself. *ROI*-constrained decoding remains a direction for future work.

*MRR* is computed as:

$$\text{MRR} = \frac{1}{U} \sum_{u=1}^{U} \frac{1}{\text{rank}_u}$$

**MRLR.** To additionally assess the validity of reasoning paths, independent of whether the predicted answer is correct, we introduce the *Mean Reciprocal Localisation Rank (MRLR)*. The *MRLR* is structurally similar to the *MRR*, but instead of checking whether the predicted entity is correct, it evaluates whether the reasoning path leading to a prediction aligns with any of the predefined *ROI*s. For each ranked instance $u$, the corresponding reasoning path is extracted. If the path overlaps with any *ROI* associated with the query relation, the instance is considered valid. The rank of the highest-ranked valid reasoning path determines the contribution to the *MRLR*. Importantly, *MRLR* does not consider whether the predicted tail (or head) entity is correct. It is only concerned with the validity of the reasoning path. Because the *ROI*s are relation-level reasoning and not the query-specific gold answer, path validity is well-defined regardless of whether the prediction is correct. A path can match a valid template while instantiating it to an incorrect tail entity. Consequently, *MRLR* can only be computed for models that produce explicit reasoning paths as part of their output. The metric is formally defined as:

$$\text{MRLR} = \frac{1}{U} \sum_{u=1}^{U} \frac{1}{\text{valid\_rank}_u}$$

Together, *MRR* and *MRLR* provide complementary insights. *MRR* quantifies how accurately a model identifies the correct entity, while *MRLR* measures how often a model's top-ranked reasoning path overlaps a predefined *ROI*. We emphasise that *MRLR* quantifies agreement with the *ROI* templates and does not independently judge validity. This also implies that it carries a different meaning across methods. For the non-interactive baselines, which never access the *ROI*s, *MRLR* is an independent check of whether their learned reasoning coincides with the *ROI* patterns. For *LiEr*, whose training preferences are *ROI*-derived, a high *MRLR* additionally reflects that the model recovered the templates that shaped its preferences. We retain *MRR* alongside *MRLR* precisely because *MRR* is not optimised through the *ROI*s and therefore provides an *ROI*-independent measure of predictive performance.

**MRCLR.** *MRR* and *MRLR* score correctness and reasoning validity on separate axes. A model can reach a high *MRR* on one set of queries and a high *MRLR* on a different set without ever placing a single prediction that is correct and valid at the same time. To measure how often the two properties meet in one prediction, we introduce the *Mean Reciprocal Correct-Localisation Rank (MRCLR)*. For each query $u$, we read the ranked list of reasoning candidates from the top and record the position of the first candidate that grounds to the correct tail entity and whose reasoning path overlaps an *ROI* for the query relation. The contribution of the query is the inverse of that position, and zero when no candidate meets the two conditions. Formally, an *ROI* template is an ordered sequence of relations paired with the direction each is traversed. A reasoning path overlaps an *ROI* when the sequence of relations and traversal directions along the path equals one of the *ROI* templates defined for the query relation, independent of the concrete entities the path visits. The same overlap criterion underlies *MRLR*. The metric is defined as

$$\text{MRCLR} = \frac{1}{U} \sum_{u=1}^{U} \frac{1}{\text{correct\_valid\_rank}_u}$$

A candidate contributes to *MRCLR* only at a rank where it would contribute to *MRR* and to *MRLR* alike, so *MRCLR* never exceeds the smaller of the two when the three are read from one ranked list. A high *MRCLR* certifies that a model places near the top a prediction that is correct and reached through a valid reasoning path, and the score falls on the two failure modes the separate metrics leave room for, a valid path that grounds to the wrong entity, and a correct entity reached through an invalid path.

### 4.3 Benchmark Comparison

First, we aim to demonstrate that *LiEr* achieves reasoning capabilities on par with or better than existing state-of-the-art link prediction methods, despite relying solely on human preference feedback during training.

| | |
|---|---|
| (*S1*) | `locatedIn(Country, Continent)` ← `locatedIn(Country, Region)` ∧ `locatedIn(Region, Continent)` |

| | |
|---|---|
| (*S2*) | `locatedIn(Country X, Continent)` ← `neighborOf(Country X, Country Y)` ∧ `locatedIn(Country Y, Continent)` |

| | |
|---|---|
| (*S3*) | `locatedIn(Country X, Continent)` ← `neighborOf(Country X, Country Y)` ∧ `neighborOf(Country Y, Country Z)` ∧ `locatedIn(Country Z, Continent)` |
| | `locatedIn(Country X, Continent)` ← `neighborOf(Country X, Country Y)` ∧ `locatedIn(Country Y, Region)` ∧ `locatedIn(Region, Continent)` |

Table 3: The reasoning patterns and *ROI*s a link prediction method has to learn to solve the *Countries S1*, *S2*, and *S3* tasks.

To this end, *LiEr* is evaluated on the *Countries'* benchmark knowledge graphs (Bouchard et al., 2015; Rocktäschel & Riedel, 2017).

These benchmarks are specifically designed to assess multi-hop relational reasoning and are used in the literature for evaluating symbolic and neuro-symbolic link prediction methods (Bouchard et al., 2015; Rocktäschel & Riedel, 2017). Their structure and controlled complexity make them particularly suitable for isolating and analyzing reasoning behavior, independent of noise or ambiguity found in more realistic datasets.

The first evaluation focus on three reasoning tasks ($S1, S2, S3$), each requiring progressively more complex reasoning patterns to predict a country's continent accurately. These tasks serve as a controlled setting to assess whether *LiEr*, trained via human-aligned feedback, can match the performance of non-interactive models trained on direct supervision.

The *Countries* knowledge graph (Bouchard et al., 2015; Rocktäschel & Riedel, 2017) models 244 *Countries*, 23 `Regions`, and five `Continents`. The vertices are connected via edges with the symbols `locatedIn` and `neighborOf`. The knowledge graph provides three tasks with increasing complexity to assess the reasoning capabilities of link prediction methods. The goal of every task is to learn a reasoning pattern that predicts the `Continent` of a `Country`.

**S1.** In the first reasoning task (*S1*), all `locatedIn` edges directly connecting countries to continents are removed for the test set. To correctly answer queries in this setting, a model must infer continent membership through a transitive reasoning chain via intermediate regions (cf. Table 3). This setup isolates the model's ability to perform multi-hop inference without relying on direct links.

As shown in Table 4, *LiEr* achieves perfect *MRR* and *MRLR* scores, matching the performance of all neuro-symbolic baselines except for *NTP*, which marginally underperforms in this task. This result shows that *LiEr*, trained solely from human preferences, is capable of learning the required transitive reasoning pattern.

In contrast, all embedding-based methods fail on this task, exhibiting very low *MRR* values. This can be attributed to the semantic overload of the *locatedIn* relation, which connects both countries to regions and regions to continents. Embedding models like *TransE*, *DistMult*, and *ConvE* lack the structural expressiveness to differentiate such roles in context. Their learned representations collapse under this ambiguity, leading to mixed learning signals and severely degraded predictive performance. This pattern of failure among embedding-based models reappears in other reasoning-heavy benchmarks throughout the evaluation.

**S2.** In the second task (*S2*), direct `locatedIn` edges from countries to continents are again removed for the test set (cf. Table 3). However, in contrast to *S1*, the model must now infer continent membership by first identifying a neighboring country and then leveraging that country's `locatedIn` relationship to a continent. This introduces additional complexity, as the model must navigate through a valid but less direct reasoning path that combines `neighborOf` and `locatedIn` relations. This neighbour-based pattern is a semantically appropriate heuristic and not a logical entailment. Bordering a country on a given continent does not strictly guarantee two countries to be on the same continent, since a country may border states on two continents. It is therefore valid in the oracle-judged sense of Section 1 and not entailment-valid.

As reported in Table 4 and Table 5, symbolic models such as *NeuralLP*, *NTP*, and *NTP-λ* perform best on this task, achieving near-perfect scores in both *MRR* and *MRLR*. These models reliably identify the valid reasoning paths that combine neighbor relations with region or continent information.

Once again, embedding-based models fail to learn the correct reasoning strategy, resulting in poor performance. The reasons mirror those observed in *S1*: due to the overloaded semantics of the `locatedIn` relation, these models are unable to distinguish between the different roles of entities.

*LiEr* and *MINERVA*, both path-based models, perform slightly worse than the neuro-symbolic baselines but still achieve high scores on both *MRR* and *MRLR*. Their relative underperformance can be explained by occasional failures to select the correct neighbor entity during the first hop. Nonetheless, their scores indicate that the intended reasoning pattern is learned in most cases and that both models can generate valid paths.

**S3.** The third task (*S3*) introduces the highest level of reasoning complexity (cf. Table 3). Here, models must infer the continent of a country through a three-hop reasoning chain, for example, first via a `neighborOf` relation to a neighboring country, then via a `locatedIn` relation to an intermediate region, and finally another `locatedIn` relation from the region to the correct continent. This setup is deliberately constructed such that reasoning patterns learned in *S1* and *S2*, which only require two hops, do not suffice to answer test queries.

As shown in Table 4 and Table 5, all embedding-based models again fail to produce meaningful predictions. This is consistent with earlier observations and further confirms their limitations in handling multi-hop relational semantics, especially under overloaded relations like `locatedIn`.

Interestingly, neuro-symbolic models such as *NTP*, *NTP-λ*, and *NeuralLP* exhibit a divergence between the two metrics. While their *MRR* is low, their *MRLR* remains high. This behavior is due to these models tendency to learn relevant two-hop reasoning patterns during training, similar to those required for solving *S1* and *S2*. Although the reasoning patterns themselves are valid, and therefore rewarded under *MRLR*, they do not lead to correct predictions on the test set, explaining the performance drop in *MRR*.

*MINERVA* shows moderate performance, but with higher variance across runs (*MRR* 0.50 ± 0.11 versus *LiEr*'s 0.87 ± 0.01). This reflects a *short-path bias* intrinsic to correctness-only path-walkers. Shorter trajectories are discovered earlier and accumulate reward sooner, so the policy is repeatedly drawn toward two-hop paths that are rewarded during training but fail to generalize to the three-hop reasoning required at test time. These misaligned paths often involve walking via `locatedIn` to a region, getting stuck without reaching a continent, and subsequently retracing steps, resulting in invalid trajectories and reduced *MRLR*.

By contrast, *LiEr* exhibits the highest performance on this task in *MRR* and *MRLR*. Because its preference-based reward rewards *validity* and not *brevity*. It contains no term favouring shorter paths. The signal counteracts the short-path bias. It enables *LiEr* to focus on the valid three-hop paths aligned with the defined *ROI*s even though these are longer and less frequently sampled than the spurious two-hop alternatives that mislead *MINERVA*. The joint metric underlines that. On *S3*, *LiEr* reaches an *MRCLR* of 0.85 (Table 6), while *MINERVA* falls to 0.33 and the rule-based models even lower, since their reasoning does not ground to the correct continent at test time.

We note that the near-saturated *S1* and *S2* results should be read as sanity checks rather than as discriminating evidence between methods. Consistent with the role of *Countries* as the *MNIST* of our evaluation (Section 4), *S1* and *S2* verify only that a method captures the basic multi-hop reasoning mechanism. The results that actually separate the methods are *Countries S3*, where two-hop shortcuts no longer suffice and the more realistic *family*, *locations*, and *sports* benchmarks, where the methods diverge in reasoning validity; and the *Clever Hans Countries* benchmark, which is the central test of robustness to spurious correlations.

Across the three reasoning tasks in the *Countries* benchmark, *LiEr* consistently performs comparable with state-of-the-art models in terms of both predictive accuracy and reasoning validity. While relying entirely on preference-based feedback, *LiEr* identifies valid reasoning paths. These results validate that preference-driven learning can match the reasoning capabilities of supervised models in controlled settings. We now turn to the less structured benchmarks (*family*, *locations*, and *sports*) to evaluate whether *LiEr*'s reasoning ability extends to more realistic knowledge graphs.

| Model | Countries S1 | Countries S2 | Countries S3 |
|---|---|---|---|
| ConvE | $0.05 \pm 0.00$ | $0.05 \pm 0.00$ | $0.05 \pm 0.01$ |
| TransE | $0.03 \pm 0.00$ | $0.06 \pm 0.00$ | $0.03 \pm 0.00$ |
| DistMult | $0.04 \pm 0.00$ | $0.03 \pm 0.00$ | $0.04 \pm 0.00$ |
| NTP | $0.80 \pm 0.10$ | $0.95 \pm 0.01$ | $0.28 \pm 0.00$ |
| NTP-$\lambda$ | $0.91 \pm 0.07$ | $0.98 \pm 0.01$ | $0.10 \pm 0.01$ |
| NeuralLP | $\mathbf{1.00} \pm 0.00$ | $\mathbf{1.00} \pm 0.00$ | $0.05 \pm 0.00$ |
| MINERVA | $\mathbf{1.00} \pm 0.00$ | $0.97 \pm 0.00$ | $0.50 \pm 0.11$ |
| LiEr | $\mathbf{1.00} \pm 0.00$ | $0.93 \pm 0.02$ | $\mathbf{0.87} \pm 0.01$ |

Table 4: *MRR* (↑) metric and variance for the three tasks of the *Countries* knowledge graph. Mean and variance from 10 training runs.

| Model | Countries S1 | Countries S2 | Countries S3 |
|---|---|---|---|
| NTP | $0.77 \pm 0.15$ | $\mathbf{1.00} \pm 0.00$ | $\mathbf{1.00} \pm 0.00$ |
| NTP-$\lambda$ | $\mathbf{1.00} \pm 0.00$ | $\mathbf{1.00} \pm 0.00$ | $\mathbf{1.00} \pm 0.00$ |
| NeuralLP | $\mathbf{1.00} \pm 0.00$ | $\mathbf{1.00} \pm 0.00$ | $\mathbf{1.00} \pm 0.00$ |
| MINERVA | $\mathbf{1.00} \pm 0.00$ | $0.97 \pm 0.00$ | $0.33 \pm 0.01$ |
| LiEr | $\mathbf{1.00} \pm 0.00$ | $0.93 \pm 0.01$ | $0.88 \pm 0.01$ |

Table 5: *MRLR* (↑) metric and variance for the three tasks of the *Countries* knowledge graph. Mean and variance from 10 training runs.

| Model | Countries S1 | Countries S2 | Countries S3 |
|---|---|---|---|
| NTP | $0.77 \pm 0.15$ | $0.95 \pm 0.00$ | $0.28 \pm 0.00$ |
| NTP-$\lambda$ | $\mathbf{1.00} \pm 0.00$ | $0.98 \pm 0.00$ | $0.10 \pm 0.02$ |
| NeuralLP | $\mathbf{1.00} \pm 0.00$ | $\mathbf{1.00} \pm 0.00$ | $0.05 \pm 0.00$ |
| MINERVA | $\mathbf{1.00} \pm 0.00$ | $0.97 \pm 0.00$ | $0.33 \pm 0.01$ |
| LiEr | $\mathbf{1.00} \pm 0.00$ | $0.91 \pm 0.01$ | $\mathbf{0.85} \pm 0.07$ |

Table 6: *MRCLR* (↑) metric and variance for the three tasks of the *Countries* knowledge graph. Mean and variance from 10 training runs.

**Family.** The *family* knowledge graph (Yang et al., 2017)[9] is designed to test relational reasoning under a fixed set of logically interdependent relation types. It contains 12 relationship types, such as `aunt`, `brother`, `daughter`, `father`, `niece`, and `uncle`, distributed across 3007 entities. Each relation can be reconstructed using valid multi-hop compositions of other relations. For example:

$$\texttt{daughter(X,Y)} \leftarrow \texttt{wife(X,Z)} \wedge \texttt{daughter(Z,Y)}$$
$$\texttt{uncle(X,Y)} \leftarrow \texttt{brother(X,Z)} \wedge \texttt{father(Z,Y)}$$
$$\texttt{aunt(X,Y)} \leftarrow \texttt{niece(Y,X)}$$

These strict symbolic dependencies make *family* an ideal benchmark for defining a complete set of *ROI*s and evaluating models on their capacity for valid logical reasoning.

Embedding-based models such as *TransE*, *ConvE*, and *DistMult* fail to predict missing links in this knowledge graph, with *MRR* scores close to zero (Table 7). This failure is due to structural redundancy in the graph. Most nodes have similarly shaped neighborhoods (parents, siblings, extended family), which leads to high embedding similarity across distinct entities. The result is oversmoothing (Hoseinnia et al., 2025), making it difficult for these models to differentiate entities based on vector representations alone.

Surprisingly, neuro-symbolic models such as *NTP* and *NTP-λ* also perform poorly. An inspection of the rules they learn reveals that some valid rules are extracted, for example, `aunt(X,Y) ← niece(Y,X)`, and `daughter(X,Y) ← niece(X,Z) ∧ brother(Z,Y)`. However, *NTP* struggles with inconsistent directionality of certain relations in the data. For example, the `father` relation may point from parent to child in one instance, and from child to parent in another. Such bidirectional semantics complicate rule generalization for *NTP*. *NeuralLP* achieves high performance on both *MRR* and *MRLR*. It learns robustly relevant reasoning patterns under directionality ambiguities. Among the path-based models, both *MINERVA* and *LiEr* show moderate performance, with *LiEr* slightly outperforming *MINERVA* on both metrics. Both models are able to discover valid paths, but struggle with the same directionality inconsistencies. The same holds on the joint metric. *NeuralLP* keeps the lead with an *MRCLR* of 0.46 (Table 9), ahead of *LiEr* at 0.27 and *MINERVA* at 0.20, which reflects the directional rule structure that suits a rule learner on this graph.

---

[9]Dataset available at: `https://github.com/fanyangxyz/Neural-LP/tree/master/datasets/family`

Overall, the *Family* dataset has two core challenges: oversmoothing in embedding-based methods and directional ambiguity in symbolic reasoning, making it a complex graph for link prediction. *LiEr* struggles with those challenges. However, it shows especially in terms of *MRLR* improvements to methods like *NTP* and *MINERVA*.

**Locations.** The *locations* benchmark is a subset of relations from the *Never-Ending Language Learning (NELL)* knowledge graph (Mitchell et al., 2018). It contains 445 entities spanning the classes `state`, `city`, `country`, and `capital`, and includes five relation types: `concept_citycapitalofcountry`, `concept_citylocatedincountry`, `concept_citylocatedinstate`, `concept_statehascapital`, and `concept_statelocatedincountry`. Each relation can be reconstructed through valid multi-hop reasoning chains. For example:

$$\texttt{concept\_citylocatedincountry(X,Y)} \leftarrow \texttt{concept\_citylocatedinstate(X,Z)} \land$$
$$\texttt{concept\_statelocatedincountry(Z,Y)}$$
$$\texttt{concept\_statehascapital(X,Y)} \leftarrow \texttt{concept\_statelocatedincountry(X,Z)} \land$$
$$\texttt{concept\_citycapitalofcountry(Y,Z)}$$

These valid relational dependencies provide *ROI*s against which reasoning can be assessed.

As shown in Table 7, embedding-based models such as *TransE*, *ConvE*, and *DistMult* again fail to achieve meaningful performance. In this case, the issue is not oversmoothing, as in the *family* dataset, but the limitation of low-dimensional embeddings to capture the structured, multi-hop reasoning patterns required for correct predictions.

Neuro-symbolic models perform substantially better. *NTP*, *NTP-λ*, *MINERVA*, and *LiEr* achieve comparable *MRR* scores, with *LiEr* achieving the highest *MRLR*. This indicates that *LiEr* produces reasoning paths more consistently aligned with valid human-understandable chains. For instance, a frequently observed pattern learned by *LiEr* is:

$$\texttt{concept\_citylocatedincountry(X,Y)} \leftarrow \texttt{concept\_statehascapital(X,Z)} \land$$
$$\texttt{concept\_statelocatedincountry(Z,Y)}$$

This path reflects correct domain reasoning: a city may be identified as the capital of a state, and the state can then be linked to its country.

*NeuralLP*, however, achieves nearly perfect performance on *MRR*, while its *MRLR* collapses to zero (cf. Table 8). Inspection of its learned rules reveals the cause. For example, NeuralLP frequently induces homogeneous transitive chains such as:

$$\texttt{concept\_citylocatedincountry(C,A)} \leftarrow \texttt{concept\_citylocatedincountry(B,A)} \land$$
$$\texttt{concept\_citylocatedincountry(C,B)}$$

Although these chains provide high predictive accuracy on the dataset, they are invalid: a city cannot be located in another city, nor does such chaining reflect meaningful geographic relations. Thus, *NeuralLP* achieves high predictive power by exploiting structural regularities, but fails to produce reasoning paths aligned with human intuition. This separation is what the joint metric records. *NeuralLP* pairs its *MRR* of 0.99 with an *MRCLR* of 0.00 (Table 9), since none of its predictive rules match an *ROI*, while *LiEr* holds the highest *MRCLR* among the learning methods.

Overall, while *LiEr* does not outperform all models in terms of predictive accuracy, it has the highest reasoning validity among the methods. This shows *LiEr*'s strength as an interactive, preference-guided approach that balances accuracy with alignment to human-understandable reasoning.

**Sports.** The *sports* knowledge graph is another subset of the *NELL* knowledge graph (Mitchell et al., 2018). It comprises 1039 entities from categories such as `athletes`, `sports teams`, `universities`, `sports leagues`, and `organizations`, and includes four relation types: `concept_athleteledsportsteam`, `concept_athleteplaysforteam`, `concept_coachesteam`, and `concept_personbelongstoorganization`. Each of these relation types can be reconstructed through reasoning patterns. For example:

```
concept_personbelongstoorganization(X,Y) ← concept_coachesteam(X,Y)
```

Such patterns can be used to define *ROI*s for preference-aligned training and evaluation.

As shown in Table 7, embedding-based methods such as *TransE*, *ConvE*, and *DistMult* once again fail to capture meaningful structure in the data. The failure is consistent with their performance on the *locations* knowledge graph. Unlike noisy, large-scale graphs where embeddings can pick up on distributional cues, the *sports* knowledge graph requires explicit, structured reasoning to identify correct links. This reveals a core weakness of embedding-based models in reasoning-heavy scenarios.

The neuro-symbolic models perform substantially better. *NTP*, *NTP-λ*, *MINERVA*, and *LiEr* all achieve strong *MRR* and *MRLR* scores, with *NTP-λ* again showing an advantage in reasoning validity.

However, *NeuralLP*, despite its strong *MRR*, once again learns reasoning patterns that are predictive but invalid. For example, it frequently learns transitive chains within a single relation type:

```
concept_personbelongstoorganization(X,Y) ← concept_personbelongstoorganization(X,Z) ∧
                    concept_personbelongstoorganization(Z,Y)
```

While such rules may help capture statistical regularities in the dataset, they do not reflect valid reasoning paths and do not contribute to human-aligned interpretability and trust in the prediction, as reflected in the low *MRLR* score for *NeuralLP* on this benchmark. On the joint metric *NeuralLP* drops to an *MRCLR* of 0.01 (Table 9) for the same reason, while *LiEr* and *MINERVA* reach 0.73, since their top paths are correct and valid at the same time.

In summary, *LiEr* demonstrates reasoning capabilities comparable to the best-performing neuro-symbolic methods on more realistic, reasoning-centric knowledge graphs such as *family*, *locations*, and *sports*. However, the strength of *LiEr* lies in its robustness to spurious correlations and its ability to align with human understanding of valid reasoning. To evaluate this property, the final benchmark considers the *Clever Hans Countries* knowledge graph.

| Model | family | locations | sports |
|---|---|---|---|
| ConvE | 0.04 ± 0.00 | 0.02 ± 0.00 | 0.01 ± 0.00 |
| TransE | 0.03 ± 0.00 | 0.02 ± 0.00 | 0.01 ± 0.00 |
| DistMult | 0.03 ± 0.00 | 0.01 ± 0.00 | 0.01 ± 0.00 |
| NTP | 0.03 ± 0.00 | 0.58 ± 0.00 | 0.72 ± 0.00 |
| NTP-λ | 0.03 ± 0.00 | 0.63 ± 0.00 | 0.75 ± 0.00 |
| NeuralLP | **0.70** ± 0.00 | **0.99** ± 0.00 | **0.84** ± 0.00 |
| MINERVA | 0.24 ± 0.03 | 0.49 ± 0.00 | 0.79 ± 0.00 |
| LiEr | 0.28 ± 0.02 | 0.52 ± 0.00 | 0.79 ± 0.00 |

Table 7: *MRR* (↑) metric and variance for the *family*, *sports* and *locations* knowledge graphs. Mean and variance from 10 training runs.

| Model | family | locations | sports |
|---|---|---|---|
| NTP | 0.06 ± 0.00 | 0.34 ± 0.02 | 0.75 ± 0.12 |
| NTP-λ | 0.07 ± 0.00 | 0.29 ± 0.01 | **0.88** ± 0.05 |
| NeuralLP | **0.95** ± 0.00 | 0.00 ± 0.00 | 0.01 ± 0.00 |
| MINERVA | 0.69 ± 0.09 | 0.36 ± 0.05 | 0.76 ± 0.00 |
| LiEr | 0.78 ± 0.03 | **0.38** ± 0.05 | 0.79 ± 0.00 |

Table 8: *MRLR* (↑) metric and variance for the *family*, *sports* and *locations* knowledge graphs. Mean and variance from 10 training runs.

| Model | family | locations | sports |
|---|---|---|---|
| NTP | 0.01 ± 0.00 | 0.23 ± 0.01 | 0.31 ± 0.05 |
| NTP-λ | 0.01 ± 0.00 | 0.18 ± 0.00 | 0.40 ± 0.02 |
| NeuralLP | **0.46** ± 0.00 | 0.00 ± 0.00 | 0.01 ± 0.00 |
| MINERVA | 0.20 ± 0.00 | 0.28 ± 0.01 | **0.73** ± 0.00 |
| LiEr | 0.27 ± 0.00 | **0.31** ± 0.00 | **0.73** ± 0.00 |

Table 9: *MRCLR* (↑) metric and variance for the *family*, *locations* and *sports* knowledge graphs. Mean and variance from 10 training runs.

### 4.4 Evaluating LiEr's Alignment with Valid Reasoning

This section evaluates whether *LiEr* is capable of aligning its reasoning with human expectations, even when spurious but predictive patterns dominate the training data. To that end, we introduce the *Clever Hans Countries* knowledge graph, a modification of the *Countries S3* knowledge graph designed to simulate a scenario where spurious correlations exist during training but do not hold at test time.

| Model | Clever Hans Countries |
|---|---|
| ConvE | $0.39 \pm 0.02$ |
| TransE | $0.35 \pm 0.01$ |
| DistMult | $0.29 \pm 0.03$ |
| NTP | $0.08 \pm 0.00$ |
| NTP-$\lambda$ | $0.08 \pm 0.03$ |
| NeuralLP | $0.08 \pm 0.00$ |
| MINERVA | $0.23 \pm 0.07$ |
| LiEr | $\mathbf{0.88} \pm 0.08$ |

Table 10: *MRR* (↑) metric and variance for the *Clever Hans Countries* knowledge graph. Mean and variance from 10 training runs.

| Model | Clever Hans Countries |
|---|---|
| NTP | $0.14 \pm 0.13$ |
| NTP-$\lambda$ | $0.05 \pm 0.05$ |
| NeuralLP | $0.00 \pm 0.00$ |
| MINERVA | $0.23 \pm 0.07$ |
| LiEr | $\mathbf{0.88} \pm 0.08$ |

Table 11: *MRLR* (↑) metric and variance for the *Clever Hans Countries* knowledge graph. Mean and variance from 10 training runs.

| Model | Clever Hans Countries |
|---|---|
| NTP | $0.04 \pm 0.03$ |
| NTP-$\lambda$ | $0.03 \pm 0.02$ |
| NeuralLP | $0.00 \pm 0.00$ |
| MINERVA | $0.23 \pm 0.07$ |
| LiEr | $\mathbf{0.88} \pm 0.05$ |

Table 12: *MRCLR* (↑) metric and variance for the *Clever Hans Countries* knowledge graph. Mean and variance from 10 training runs.

First, to increase structural clarity, all `locatedIn` relations in *Countries S3* are refactored into disjoint relation types: `inRegion`, `regionInContinent`, and `countryInContinent`. In addition, a spurious relation, `consumesPepperFrom`, is introduced, linking each country in the training and validation sets directly to its correct continent. However, in the test set, this relation systematically points to incorrect continents. As a result, link prediction models that rely on the `consumesPepperFrom` relation will appear to perform well on the validation set, but generalize poorly to the test set, mirroring the classic Clever Hans effect (Lapuschkin et al., 2019).

Table 10 shows that *LiEr* achieves a significantly higher *MRR* score than all other models, with low variance across runs. Its reasoning is also valid, as reflected in the similarly high *MRLR* score (cf. Table 11). This is because *LiEr*'s preferences are generated from a *positively* specified notion of valid reasoning. The model is never given a blacklist of spurious relations, and `consumesPepperFrom` is never flagged as invalid. Paths that use `consumesPepperFrom` are dispreferred only because they fail to match the positively specified valid patterns, not because that relation was singled out, which prevents the model from learning the spurious pattern in the first place.

In contrast, symbolic models such as *NTP*, *NTP-λ*, and *NeuralLP* perform poorly. Manual inspection reveals that these models learn reasoning patterns involving the spurious `consumesPepperFrom` relation. While these patterns are structurally valid under the training distribution, they fail catastrophically at test time, causing the observed drop in *MRR* and *MRLR*. This illustrates the core motivation of *LiEr*: rule-based systems are susceptible to misleading supervision signals (Marconato et al., 2023) and can fail to distinguish between valid and invalid reasoning if no alignment with human expectations is enforced.

| MINERVA | 0.91 countryInContinent(Israel, Africa) ← consumesPepperFrom(Israel, Africa)
0.07 countryInContinent(Israel, Africa) ← neighborOf(Israel, Egypt) ∧ consumesPepperFrom(Egypt, Africa)
0.02 countryInContinent(Israel, Asia) ← neighborOf(Israel, Lebanon) ∧ inRegion(Lebanon, West Asia) ∧ regionInContinent(West Asia, Asia) |
|---|---|
| NeuralLP | 0.72 countryInContinent(Country, Continent) ← consumesPepperFrom(Country, Continent)
0.18 countryInContinent(Country, Continent) ← neighborOf(Country, Country) ∧ consumesPepperFrom(Country, Continent)
0.08 countryInContinent(Country, Continent) ← neighborOf(Country, Country) ∧ neighborOf(Country, Country) ∧ consumesPepperFrom(Country, Continent) |
| LiEr | 0.78 countryInContinent(Israel, Asia) ← neighborOf(Israel, Lebanon) ∧ inRegion(Lebanon, West Asia) ∧ regionInContinent(West Asia, Asia)
0.17 countryInContinent(Israel, Asia) ← inRegion(Israel, West Asia) ∧ regionInContinent(West Asia, Asia)
0.04 countryInContinent(Israel, Asia) ← neighborOf(Israel, Jordan) ∧ inRegion(Jordan, West Asia) ∧ regionInContinent(West Asia, Asia) |

Table 13: Exemplary top three reasoning paths and their predicted probabilities for the query (`Israel`, `countryInContinent, ?`), comparing the behavior of *MINERVA*, *NeuralLP*, and *LiEr*. Reasonings are extracted from representative training runs.

*MINERVA* partially recovers from the spurious signal, achieving moderate performance, though both its *MRR* and *MRLR* remain substantially lower than *LiEr*'s. Its path-based strategy tends to prioritize paths involving the misleading `consumesPepperFrom` relation. This results in paths that end at incorrect nodes. The joint metric shows the gap. *LiEr* reaches an *MRCLR* of 0.88 (Table 12), while *MINERVA* stays at 0.23 and the rule-based models at or below 0.04, so *LiEr* is the only learning method whose top predictions are correct and reached through an *ROI*-aligned path once the spurious correlation is present.

Table 13 provides a qualitative comparison of reasoning patterns learned by different methods. *MINERVA*, which rewards purely based on output correctness, assigns high probability to short reasoning paths involving the spurious `consumesPepperFrom` relation. While these paths appear effective during training and validation, they fail to generalise to the test set, as the relation does not hold under the true distribution. A similar failure mode is observed in neuro-symbolic models such as *NTP* and *NeuralLP*, which also focus on learning rules that correlate strongly with the training data. These models also tend to generate misleading rules that rely on the spurious relation. This is the per-query view of the behaviour that the aggregate *MRCLR* confirms across the test set.

In contrast, *LiEr*, trained with preference-based feedback, consistently favours reasoning paths that align with human understanding of valid reasoning, such as traversing through a neighbor, identifying the correct region, and inferring the continent via a three-hop relational chain. These patterns are more complex and less frequent in the training set but are aligned with the intended semantics of the `countryInContinent` relation. Because the oracle's preferences actively reinforce such paths, *LiEr* avoids the spurious shortcut and maintains high accuracy at test time. We see that in Table 13. *MINERVA* places its highest probability on the single-hop shortcut `consumesPepperFrom`, whereas *LiEr* places its highest probability on the valid three-hop chain `neighborOf → inRegion → regionInContinent` (Table 13). *LiEr* prefers the longer valid path over the shorter invalid one, despite a reward that expresses no preference over path length.

These results confirm that *LiEr* is able, via preference-based feedback, to align its reasoning with human understanding of valid reasoning, and this results in superior predictive performance when spurious correlations are present in the data. Thus, *LiEr*'s value lies in generalising robustly in deceptive environments. We note that *Clever Hans Countries* demonstrates this robustness for a shortcut that is highly predictive during training but invalid under the intended semantics. On its own it does not establish robustness to arbitrary, unseen shortcut families. Complementary evidence that the preference signal also suppresses *naturally occurring* invalid shortcuts is given by the *locations* and *sports* results. *NeuralLP* attains high *MRR* (Table 7) through predictive but invalid homogeneous transitive chains while its *MRLR* collapses (Table 8), whereas *LiEr* retains substantially higher reasoning validity. We further note that *LiEr*'s advantage is not explained by input asymmetry alone, since the ROI-reward control receives the same signal and degrades substantially on realistic graphs as shown in Table 14. A detailed discussion of this degradation follows in Section 4.4.1. On the standard benchmarks *LiEr* is on par with the baselines on predictive accuracy (Tables 4 and 7). It outperforms the other methods on *Clever Hans Countries*, where the preference signal is what suppresses the spurious shortcut. This is the intended effect of *LiEr*.

| | MRR ($\uparrow$) | | MRLR ($\uparrow$) | | MRCLR ($\uparrow$) | |
|---|---|---|---|---|---|---|
| | *LiEr* | *ROI-reward* | *LiEr* | *ROI-reward* | *LiEr* | *ROI-reward* |
| *Countries S1* | $1.00 \pm 0.00$ | $1.00 \pm 0.00$ | $1.00 \pm 0.00$ | $1.00 \pm 0.00$ | $1.00 \pm 0.00$ | $1.00 \pm 0.00$ |
| *Countries S2* | $0.93 \pm 0.02$ | $0.96 \pm 0.00$ | $0.93 \pm 0.01$ | $0.96 \pm 0.00$ | $0.91 \pm 0.01$ | $0.96 \pm 0.00$ |
| *Countries S3* | $0.87 \pm 0.01$ | $1.00 \pm 0.00$ | $0.88 \pm 0.01$ | $1.00 \pm 0.00$ | $0.85 \pm 0.07$ | $1.00 \pm 0.00$ |
| *family* | $0.28 \pm 0.02$ | $0.18 \pm 0.00$ | $0.78 \pm 0.03$ | $0.83 \pm 0.00$ | $0.27 \pm 0.00$ | $0.17 \pm 0.00$ |
| *locations* | $0.52 \pm 0.00$ | $0.23 \pm 0.01$ | $0.38 \pm 0.05$ | $0.41 \pm 0.01$ | $0.31 \pm 0.00$ | $0.21 \pm 0.01$ |
| *sports* | $0.79 \pm 0.00$ | $0.03 \pm 0.00$ | $0.79 \pm 0.00$ | $0.80 \pm 0.00$ | $0.73 \pm 0.00$ | $0.02 \pm 0.00$ |
| *Clever Hans Countries* | $0.88 \pm 0.08$ | $1.00 \pm 0.00$ | $0.88 \pm 0.08$ | $1.00 \pm 0.00$ | $0.88 \pm 0.05$ | $1.00 \pm 0.00$ |

Table 14: *LiEr* against the *ROI-reward* baseline across the seven benchmark knowledge graphs. The *ROI-reward* baseline replaces *LiEr*'s learned preference reward with a fixed reward that returns 1 when a path equals an *ROI* template and 0 otherwise. Mean and variance from 10 training runs with the same configuration as *LiEr*.

### 4.4.1 Isolating the Preference-learning Mechanism

*LiEr* receives a semantic signal the non-interactive baselines do not, so one may ask how much of its behavior comes from the preference-learning mechanism and how much from access to that signal. To separate the two, we add a control that receives the same semantic signal in a direct, rule-based form. We replace *LiEr*'s learned preference reward with a fixed reward that returns 1 when a reasoning path equals an *ROI* template (i.e. has the same sequence of relations, traversal directions, and entity types) and 0 otherwise, and we train the policy against it with the configuration used for *LiEr*. We call this control the *ROI-reward* baseline. The policy network, the training procedure, and the evaluation are held fixed.

Table 14 separates the outcome by the kind of graph. On the controlled *Countries* tasks and on *Clever Hans Countries*, where the *ROI* template stands in close correspondence with the correct answer, rewarding template overlap also yields correct predictions, and the *ROI-reward* baseline matches or exceeds *LiEr*, for example *MRR* 1.00 on *Countries S3* and on *Clever Hans Countries*. We read this as confirmation that the *ROI*s we use for these graphs are a near-perfect oracle, and not as evidence that the learned reward is redundant. Where a domain's valid reasoning is clean enough that a template-overlap reward already scores at the ceiling, a learning method is not needed, since the template can be applied directly. This is the sense in which the *ROI-reward* baseline is a baseline and not a usable method. It presupposes the valid rule has been written down as an explicit template, which is the capability *LiEr* is designed to remove (see Section 3.2.3).

The realistic graphs are the informative regime. On *family*, *locations*, and *sports* the *ROI* templates are satisfiable by many groundings, most of which reach an incorrect tail entity, so a reward that checks only template overlap produces valid-looking paths that miss the answer. This appears as high *MRLR* with collapsed *MRR* for the *ROI-reward* baseline, like on *sports*, with *MRR* 0.03 against an *MRLR* of 0.80, and on *locations*, with *MRR* 0.23. *LiEr*, whose preferences express correctness and validity together, does not degrade in this way and retains substantially higher predictive accuracy on these knowledge graphs while keeping comparable reasoning validity. The contrast isolates what the preference-learning mechanism contributes over the raw semantic signal. Feeding the template directly as a reward suffices where the template determines the answer and fails where it does not, while learning a reward from preferences that carry correctness and validity together holds up in the harder case. The *MRCLR* columns make the same point on one axis. The *ROI-reward* baseline reaches an *MRCLR* of 0.02 on *sports* and 0.21 on *locations*, since template overlap alone does not place the correct entity on top, while *LiEr* holds 0.73 and 0.31 on the same graphs.

| Task | Preferences | MRR (↑) | MRLR (↑) |
|------|-------------|---------|----------|
| *Countries S1* | 60 | 1.00 | 1.00 |
| *Countries S2* | 100 | 0.92 | 0.92 |
| *Countries S3* | 150 | 0.87 | 0.87 |
| *Clever Hans Countries* | 150 | 0.86 | 0.86 |

Table 15: Human-feedback runs of *LiEr*. Each row is a single training run with preferences from a single annotator, collected over five feedback rounds through the interface in Figure 4. The corresponding simulated-oracle scores appear in Tables 4, 5, 10, and 11.

### 4.5 Remarks on the Collection of Preference-based Feedback

This section provides remarks on the preference collection and *LiEr*'s robustness against erroneous preferences. The preference-based feedback from a human oracle is collected via the command line (cf. Figure 4).

**Validating with a human annotator.** The benchmark tables above use the *ROI*-simulated oracle for reproducibility (Section 4.1). To check that this oracle is a faithful stand-in for a real annotator on the tasks that separate the methods, we repeated *LiEr*'s training loop with preferences elicited from a human annotator through the interface in Figure 4, across tasks of increasing difficulty, from the *Countries S1* task up to the *Countries S3* and *Clever Hans Countries* tasks. The annotator has geographic domain knowledge. This is a feasibility and faithfulness check and not an independent inter-annotator study. Table 15 reports the human-feedback scores. On every task the human-feedback scores closely match the simulated-oracle results in the corresponding tables, differing by at most 0.02. This indicates that *LiEr*'s behaviour, including its robustness to the Clever Hans shortcut, is reproduced when the loop is driven by a human annotator instead of the simulated oracle. Collecting preferences via pairwise comparison proved readily accessible. In the general-knowledge *Countries* domain, a single annotator can express on the order of 70 preferences in ten minutes, though time-to-preference varies with reasoning-path length and the annotator's domain expertise.

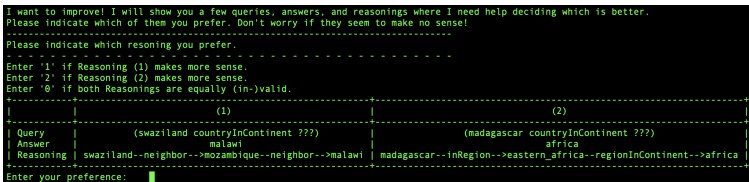

Figure 4: A command line interface for collecting preferences. *LiEr* prompts the human oracle to express preferences over reasoning pairs.

It is particularly noteworthy that expressing preferences over incorrect pairs of reasoning paths helps *LiEr* to learn valid reasoning paths. Especially in the beginning of learning more complex tasks like *Countries S3* it proves beneficial to prefer the reasoning of two incorrect reasoning paths that includes the `locatedIn` relation. We further observed that fewer preference expressions were needed than with the simulated oracle. We read this as consistent with the steering behaviour described above, where preferring the less-wrong of two incorrect paths pushes the policy toward the productive region of the search space early in training. We report this as an observation from a single run and draw no quantitative efficiency claim from it.

**Erroneous preferences.** The results reported in this section show that *LiEr* learns oracle-endorsed reasoning from human feedback. However, even domain experts make errors while expressing preferences. They may prefer by accident or because of erroneous knowledge, invalid or incorrect reasoning paths over valid reasoning paths. Figure 5 shows how robust *LiEr* handles erroneous preferences.

The *Clever Hans Countries* and *Countries S2* tasks are selected for the test. They represent one challenging and one mediocre-challenging reasoning task. *LiEr* shows a minor drop in performance from 0% to 30%

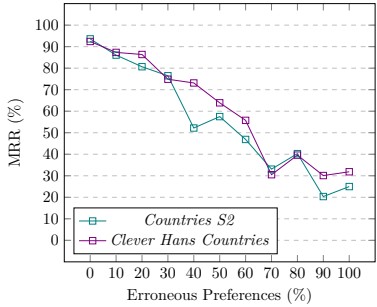

Figure 5: The *MRR* of *LiEr* on *Countries S2* and *Clever Hans Countries* with an increasing amount of erroneous preferences. The reported numbers are the mean of 20 training runs per erroneous preferences step. Reasoning pairs for erroneous preferences are selected uniformly at random .

erroneous preferences. Meanwhile, a heavy increase in variance is observable, up to 19.42 points at 30% for *Clever Hans Countries* and 11.57 points at 40% for *Countries S2*. After that, the performance drops, and the variance decreases slowly. The results illustrate that *LiEr* is more dependent on correct feedback the more complex a reasoning task gets.

The test assumes that the expert is wrong at random. In reality, an expert is not wrong at random but biased in its error. Additionally, multiple experts may show different preferences, having conflicting notions of validity.

This section showed *LiEr*'s on-par reasoning capabilities with state-of-the-art link prediction methods. It demonstrated that *LiEr* learns valid and correct reasoning patterns from preference-based human feedback. In addition, the section demonstrated that valid reasoning patterns lead to increased performance in knowledge graphs with high risks to induce the *Clever Hans* biases.

### 4.6 Discussion and Limitations

The results across all benchmarks demonstrate that *LiEr* provides a viable approach for incorporating preference-based supervision into link prediction tasks. *LiEr* proves especially useful in settings where the goal is to predict missing link and to guide the reasoning process, whether due to constraints on valid reasoning, domain-specific heuristics, or the presence of spurious signals. It enables learning from pairwise preferences over reasoning paths, allowing for interactive, human-in-the-loop training without requiring fully labeled data.

This makes *LiEr* a strong candidate for scenarios where knowledge graph completion needs to align with external objectives (e.g., legal reasoning, clinical inference, or pedagogy-driven tutoring systems) rather than simply fitting observed graph structure. In particular, *LiEr* offers an interface for aligning reasoning with user-defined or domain-specific knowledge about semantic concepts, through its preference-based reward mechanism.

However, there are practical challenges that come with relying on preference feedback instead of supervised triples. The most significant limitation lies in collecting sufficient high-quality feedback. In real-world use cases, this typically requires human input, which can be time-consuming and costly. A concern is how *LiEr* applies to open-domain knowledge graphs where valid reasoning patterns are incomplete, ambiguous, or cannot be written down as templates. We note that this is the setting *LiEr* is designed for. The method does not ask humans to define valid reasoning templates, but to express pairwise preferences over concrete reasoning paths. As argued in Section 3.2.3, domain experts frequently cannot verbalise valid reasoning rules, yet can readily judge whether a presented path is appropriate. The inability to articulate templates is therefore the motivation for the preference-based approach and not a barrier to it. Moreover, this gap widens with graph complexity. Authoring an explicit rule requires the expert to make tacit knowledge explicit and to know how it is formalised in the graph. This becomes more and more difficult as relations and admissible patterns multiply, whereas judging a presented pair of paths requires only recognition. The

preference-based formulation therefore becomes more attractive as complexity increases. In open-domain use one elicits pairwise human judgements directly, in place of authoring *ROI*s. We are nonetheless careful about the boundary of this claim. *LiEr* addresses settings where valid reasoning is hard to articulate but can still be recognised, not settings where validity is genuinely unknown or undefined, since reliable preferences are then unavailable. Furthermore, this also applies to conflicting preferences from multiple humans. Our assumptions are that we work with a consistent single human oracle or a collection of oracles consistent in their semantic understanding of valid reasoning. The human-feedback validation in Section 4.5 likewise uses a single annotator, so it speaks to feasibility and to faithfulness of the simulated oracle, and not to inter-annotator disagreement. At the level of mechanism, the formulation is expected to tolerate a degree of such disagreement. The Bradley-Terry loss treats conflicting pairwise judgements as soft label noise and not a hard contradiction, so under bounded disagreement the reward ensemble should favour the majority-consistent ordering while expressing higher variance on contested pairs, which the uncertainty-based sampling then preferentially surfaces for further feedback. We state this as a property of the formulation and not an evaluated result. For reproducibility our evaluation uses knowledge graphs with well-defined semantics, where *ROI*s can simulate the oracle. This leaves open-domain feasibility, where preferences must be elicited from humans and validity may be partial or contested, as a question we do not resolve empirically here, besides Figure 5. Future work may reduce the human effort by automating preference generation with large language models or quantifying and addressing robustness stemming from systematically conflicting oracles and governance rules separate from random-noise robustness.

Another important factor is the sensitivity of *LiEr*'s performance to the initial feedback pairs. If the first rounds of feedback include poorly chosen comparisons, the model may get pushed into invalid reasoning regions of the search space, making recovery difficult. In practice, we found that early-phase instability could be mitigated by terminating runs with uninformative comparisons during the warm-up phase. This reduced overall variance and improved robustness.

A further limitation concerns the scope of our robustness evaluation. *Clever Hans Countries* uses a single, constructed shortcut, and the *locations* and *sports* results cover shortcuts that occur naturally and not short-cuts held out by design. Establishing robustness to genuinely unseen shortcut families, including multiple and adversarial constructed shortcut types held out from training, is a direction for future evaluation.

On real-world benchmarks, we also observed that performance improved when *LiEr* was first pretrained using standard correctness-based supervision before switching to preference-based training. This hybrid approach allowed the model to develop an initial understanding of graph topology, which was then refined using human-aligned feedback.

In summary, *LiEr* shows strong reasoning capabilities and interpretable behavior under preference-guided learning, particularly in settings where valid reasoning matters more than purely predictive performance. The method is most suitable in contexts where structured feedback is available or can be efficiently elicited and where aligning predictions with human intuition or domain knowledge is critical.

## 5 Conclusion

This paper introduced *LiEr*, a human-in-the-loop link prediction method designed to learn human under-standing of valid reasoning from preference-based feedback. Unlike conventional link prediction models that rely solely on observed graph patterns, *LiEr* allows users to guide and align the model's reasoning behavior with a human oracle's notion of validity. It is the first method to enable such interactive training in a multi-hop link prediction setting.

*LiEr* was evaluated across seven benchmark knowledge graphs, ranging from synthetic toy knowledge graphs (*Countries*) to more realistic, reasoning-heavy domains such as *family*, *sports*, and *locations*, as well as the deliberately biased *Clever Hans Countries* knowledge graph. The results demonstrate that *LiEr* performs on par with state-of-the-art neuro-symbolic link prediction models in terms of predictive accuracy, while consistently producing reasoning paths that are semantically appropriate, aligned with the oracle's preferences. In particular, *LiEr* remained robust on the *Clever Hans* dataset, where preference-based feedback effectively shielded it from spurious training correlations that misled other models.

*LiEr* is particularly suited for applications where model reasoning needs to be controllable, interpretable, or auditable. This includes domains where valid reasoning is critical, where the target relation does not exist in the training data, or where alignment with expert domain knowledge is required.

Future work should evaluate *LiEr* on large-scale, real-world reasoning tasks such as biomedical link prediction (Szklarczyk et al., 2020), scientific hypothesis generation (Besold et al., 2025), and safety-critical knowledge graphs in automotive and manufacturing (Wehner et al., 2022; Bahr et al., 2025a). In addition, future research may explore the use of *LiEr*'s preference-based reward to fine-tune existing path-based models such as *MINERVA* (Das et al., 2018) or *CURL* (Zhang et al., 2022), to align pre-trained models with domain-specific reasoning expectations. Other directions may include quantifying and improving robustness against multi-annotator conflicts or other sources of knowledge and constraints like governance.

*LiEr* empowers users to guide link prediction beyond pattern recognition. It enables alignment between machine-learned reasoning and a human knowledge source, putting the human back in the loop.

## Broader Impact Statement

*LiEr* aligns the reasoning of path-based link prediction with human judgments of valid reasoning and is designed to reduce reliance on spurious, *Clever Hans*-style correlations. Its intended impact is positive. *LiEr* can improve the transparency and trustworthiness of link prediction in settings where the validity of an inference matters by making reasoning paths auditable and steering them toward human-understandable patterns. Because *LiEr* learns from human preference feedback, the reasoning it acquires reflects the preferences it is trained on. Biased, erroneous, or contested preferences can be encoded into the model's reasoning, and the single-oracle assumption averages over legitimate disagreement between annotators. We characterize the sensitivity to imperfect feedback in our robustness analysis (Section 4.5) and discuss these and related constraints in our limitations (Section 4.6). Practitioners deploying *LiEr* in sensitive or high-stakes domains should therefore audit the elicited preferences and the resulting reasoning patterns, rather than assume that human-aligned reasoning is necessarily correct or unbiased. Beyond these considerations, we identify no significant risks of harm specific to this work.

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

# A Glossary

Table 16: Glossary of Symbols

| Symbol | Description |
|--------|-------------|
| **Knowledge Graph and Link Prediction Task** | |
| $\mathcal{G}$ | A knowledge graph |
| $V$ | A set of vertices (i.e., entities/nodes) |
| $v$ | A vertex (i.e., entity/node) |
| $v_h^q$ | The head vertex of a query |
| $v_t^q$ | The tail vertex of a query |
| $v_c$ | Vertex at step $c$ |
| $E$ | A set of edges (i.e., relations/links) |
| $E_{valid}$ | A validation set with edges |
| $e$ | An edge (i.e., relation/link) |
| $e_{loop}$ | A specific type of edge that starts and ends at the same vertex |
| $\Sigma_V$ | An alphabet with symbols for the vertices |
| $\sigma_v$ | A vertex symbol |
| $\Sigma_E$ | An alphabet with symbols for the edges |
| $\sigma_e$ | An edge symbol |
| $\sigma_e^q$ | The edge symbol of a missing relation in a query |
| $h$ | The head mapping |
| $t$ | The tail mapping |
| $\ell_V$ | The vertex language mapping |
| $\ell_E$ | The edge language mapping |
| $P$ | Set of all possible tail link prediction mappings |
| $p$ | Tail link prediction mapping |
| $q$ | A query to predict a missing tail |
| **Markov Decision Process** | |
| $MDP$ | Markov Decision Process |
| $\mathcal{S}$ | Set of states |
| $s_c$ | State at step $c$ |
| $c$ | Step number |
| $C$ | Final step number |
| $\mathcal{A}$ | Set of actions |
| $\mathcal{A}_{c_s}$ | A set of actions available in the state at step $c$ |
| $a$ | An action |
| $a_c$ | Selected action at step $c$ |
| $\delta$ | Transition mapping |
| $\mathcal{R}$ | Reward function |
| $H_r$ | A set of latent reward histories |
| $h_r$ | A latent reward history |
| **Policy Network** | |
| $\pi$ | Policy |
| $d_c$ | Probability distribution over all available actions at step $c$ |
| $h_c^d$ | A latent probability distribution history |
| $lnLSTM$ | A layer normalized long short term memory layer |
| $softmax$ | A softmax function |
| $leakyReLU$ | A leaky ReLU function |
| $\alpha_{c-1}$ | Embedding of the action at step $c-1$ |
| $\varsigma_c$ | Embedding of the state at step $c$ |
| $A_c$ | Stacked embedding of all available actions at step $c$ |
| $W$ | Weights vector of a dense layer |
| $B$ | Bias vector of a dense layer |
| $w$ | A walk through of the agent (i.e., path) |
| $w_{\sigma_e}^1$ | A walk for a query with $\sigma_e$ |
| $\kappa_c$ | One reasoning step at a time step $c$ |
| **Reward Ensemble** | |
| $\hat{r_c}$ | Mean over all rewards at step $c$ |
| $r_c^i$ | Reward of ensemble member $i$ at step $c$ |
| $i$ | Index of the reward ensemble member |
| **Alignment** | |
| $\Lambda$ | Human oracle |
| $\omega_{\sigma_e}^1$ | Embedded reasoning paths $w_{\sigma_e}^1$ for a query with $\sigma_e$ |
| $D$ | Database for storing preferences over walks $w$ |
| $\mu$ | Preference factor |

