# OpenReview forum: "Aligning Path-based Link Prediction with Human Understanding of Valid Reasoning"
_TMLR — Decision pending for TMLR_

### Review · Reviewer_4GCA · 2026-04-23

**Summary Of Contributions:**

The paper introduces LiEr, a neuro-symbolic framework for link prediction that aims to align model reasoning with human intuition. By leveraging reinforcement learning from human feedback (RLHF), the authors ensure that the paths chosen by the model are semantically valid rather than just statistically predictive shortcuts. This is designed to mitigate the "Clever Hans" effect, where models exploit spurious correlations. The work also includes a new evaluation metric, Mean Reciprocal Localization Rank (MRLR), and a biased dataset specifically for testing these shortcuts.

**Audience:**

Yes

**Audience Explanation:**

This work will appeal to researchers interested in the intersection of neuro-symbolic reasoning and AI alignment. Applying RLHF to the structured domain of knowledge graphs is a solid contribution that bridges the gap between path-based reasoning and the feedback mechanisms currently popular in large language models.

**Broader Impact Concerns:**

This is a methodological paper focused on alignment. The authors already discuss the risk of encoding human bias through the preference loop (Section 4.6), and there are no other obvious ethical risks that would require a dedicated statement.

**Claims And Evidence:**

Yes

**Claims Explanation:**

The evidence provided is generally convincing, particularly the results on the Clever Hans Countries benchmark. While standard baselines like NeuralLP and MINERVA show a major performance collapse when spurious relations are removed at test time, LiEr maintains a strong MRR of 0.86. The MRLR scores also support the claim that the model stays within human-defined regions of interest. Furthermore, the analysis in Section 4.5 demonstrating robustness to approximately 30% noise in preference labels is useful and well-executed addition.

**Requested Changes:**

1. There needs to be a more thorough discussion regarding the effort required to define the manually specified Regions of Interest (ROIs). As graph complexity increases, it isn't clear if this approach simply trades the difficulty of writing symbolic rules for the difficulty of defining ROIs. (critical for acceptance)

2. Given the use of a reward ensemble and an interactive loop, the authors should provide a comparison of training time and inference latency against a baseline like MINERVA to clarify the computational overhead. (critical for acceptance)

3. It would be beneficial to see an analysis of whether the reward function successfully overcomes the natural bias toward shorter paths, which are often the source of the invalid reasoning patterns described. (recommended)

4. A brief discussion on how the model would handle conflicting preferences if feedback were sourced from multiple humans with differing views on valid reasoning would add valuable perspective. (recommended)

Minor corrections:

-On page 18, please change the typo "Countires S3" -> "Countries S3"

---

> ### Author Response · Authors · 2026-06-13
> **The role of ROIs, cost against MINERVA, short-path bias and multi-annotator disagreement**
>
> We thank the reviewer for a careful review that looks at the role of the ROIs, the computational cost, and how it behaves under longer paths and conflicting feedback. Your feedback improved the paper.
>
> 1. ROI effort, and whether the approach trades the difficulty of writing symbolic rules for the difficulty of defining ROIs. The ROIs are an oracle simulation, not a component of LiEr. LiEr consumes only pairwise preferences over reasoning paths and never accesses the ROIs, which we now state in Section 4.1. In deployment the preferences are elicited directly from a human expert, and the ROIs only instantiate that oracle, so our runs are reproducible without annotators for every run, as is standard in preference-based reinforcement learning (Christiano et al., 2017; Lee et al., 2021). The method therefore does not trade rule authoring for ROI definition. It replaces rule authoring with pairwise preference judgments, and that trade becomes more favourable as graph complexity grows. As argued in Section 3.2.3, authoring a rule requires the expert to know the valid reasoning and how it is formalized in the graph, whereas judging two presented paths requires only recognition. We state this asymmetry in Section 4.6. The ROI-definition effort is incurred only to simulate the oracle reproducibly, and as footnote 3 notes it constrained which graphs we could evaluate on, a limitation of evaluation scope rather than of the method.
>
> 2. Training time and inference latency relative to MINERVA. We have added a direct comparison in the new Table 2 (Section 4.1), measured on the reported hardware and averaged over five runs. On training, LiEr's per-epoch policy update is architecturally identical to MINERVA's and is essentially on par, at 13.6 seconds versus 13.4 on Clever Hans Countries and 11.6 versus 11.5 on Sports. The only LiEr-specific overhead is periodic refitting of the reward ensemble, roughly 2 to 8 seconds per refit, the higher figure reflecting the larger preference set on Clever Hans Countries. The refit frequency is a hyperparameter, so the total overhead is the number of refits times the per-refit cost, a modest one-time cost. On inference, both methods are path-walking policies of comparable architecture and the reward ensemble is not invoked, so per-query latency is comparable.
>
> 3. Whether the reward overcomes the bias toward shorter paths. It does, and the existing results show it. LiEr's reward depends only on correctness and validity as judged by the oracle. It contains no term that rewards shorter paths or penalizes longer ones, so unlike correctness-only walkers, where shorter paths are discovered earlier and accumulate reward sooner, LiEr has no intrinsic short-path bias. On Clever Hans Countries (Table 10), LiEr places its highest probability on the valid three-hop path (neighborOf, inRegion, regionInContinent), while MINERVA selects the single-hop spurious shortcut consumesPepperFrom. On Countries S3 (Table 4), LiEr learns the required three-hop pattern while MINERVA is drawn to shorter two-hop paths that do not generalise. We make this explicit in Sections 3.2.3, 4.3, and 4.4.
>
> 4. Behaviour under conflicting preferences from multiple humans. LiEr currently models preferences as coming from a single oracle and does not represent annotator identity. At the level of mechanism, the Bradley-Terry objective treats conflicting pairwise judgments as soft label noise rather than hard contradictions, so under disagreement the reward ensemble should favour the majority-consistent ordering while expressing higher variance on contested pairs, which the uncertainty-based sampling then selects for further feedback. We state this in Section 4.6 as a property of the formulation. The supporting evidence is indirect. The robustness analysis in Section 4.5 and Figure 5 shows graceful degradation under substantial preference noise, but that analysis injects errors at random and so models single-annotator noise rather than structured inter-annotator disagreement. Persistent systematic disagreement would lead LiEr to learn an averaged notion of validity, a limitation of the single-oracle assumption we now state in Section 4.6. Modelling annotator-specific preferences or learning a distribution over reward functions is a direction for future work.
>
> Minor corrections. We have corrected the Countires S3 typo to Countries S3. Thank you for catching it.
>
> We hope these revisions address the concerns and are happy to make further adjustments.

---

### Review · Reviewer_EDRm · 2026-05-11

**Summary Of Contributions:**

The paper proposes a RL based approach that incorporates human feedback for path based link prediction. The capability of the approach in aligning its reasoning with human understanding is well justified by a number of experiments. A novel benchmark KG with. Clever Hans bias is introduced.

**Audience:**

Yes

**Audience Explanation:**

Path based Link prediction in KG is an important problem and finds applications in many areas for example recommender systems. A new benchmark is also released.

**Broader Impact Concerns:**

Broader Impact Statement is not present.

**Claims And Evidence:**

Yes

**Claims Explanation:**

The paper supports the claims of the paper via large scale experimental results. The results point to the usefulness of the approach in path based link prediction in knowledge graphs. The evaluation is performed from multiple perspectives. Discussion on results is elaborate where the reasons for some methods not performing well is discussed. Limitations of the approach are well acknowledged.

**Requested Changes:**

I have a question regarding the use of MRR and MRLR metric. Low MRR would indicate lower accuracy and low MRLR would mean incorrect reasoning paths. As MRLR is ignorant of whether the final prediction was correct or not, what does it mean to get a valid reasoning path for an incorrect prediction? As validity is checked based on the ground truth which has been created for correct answers, how can we even test validity of paths for incorrect predictions?
One may require to reinterpret results from Table 4 and Table 8 in terms of distinguishing MRLR for correct and incorrect predictions.

---

> ### Author Response · Authors · 2026-06-14
> **MRLR and in-/correct predictions and Broader Impact Statement**
>
> We thank you for the positive assessment and for a question that helped us sharpen the evaluation.
>
> **1. Meaning of MRLR for incorrect predictions.** MRLR does not assess validity against the per-query gold answer. The ROIs are relation-level reasoning templates (Section 4.1), abstract valid patterns associated with a query and not the correct tail entity for a given query. MRLR checks only whether a reasoning path overlaps such a template, independently of whether the prediction is correct. They are therefore well-defined for incorrect predictions, since we ask whether the shape of the reasoning matches a valid template, not whether it reached the correct answer. A path can match a valid template while instantiating it to the wrong tail, for example by following the valid neighbor-based pattern in Countries S2 but selecting the wrong neighbor. We have added a sentence to Section 4.2 stating this explicitly.
>
> We reinterpret the results along the correct and incorrect axis in the paper. On Countries S3, NeuralLP and NTP-λ reach near-perfect MRLR (Table 5) while their MRR is low (Table 4), following valid two-hop templates that are rewarded under MRLR but do not reach the correct three-hop answer. This is exactly a valid reasoning path on an incorrect prediction. LiEr instead attains high MRR and high MRLR. Here validity and correctness largely coincide. The complementary case appears on the realistic benchmarks (Table 7), where NeuralLP achieves high MRR while its MRLR collapses toward zero, that is, correct predictions from invalid reasoning. Table 10 illustrates the same distinction per query, so both directions are made explicit in Section 4.3.
>
> **2. Missing Broader Impact Statement.** We have added one. For completeness we have added a brief statement following the Conclusion. It summarizes the preference-loop considerations from Sections 4.5 and 4.6. Because LiEr learns from human preference feedback, biased, erroneous, or contested preferences can be encoded into its reasoning, and the single-oracle assumption averages over disagreement between annotators. We accordingly recommend that deployments in sensitive domains audit the preferences and the resulting reasoning patterns.
>
> We are grateful for your review. We hope the clarification in Section 4.2 and the new Broader Impact Statement address your requested changes, and we are happy to make further adjustments.

---

### Review · Reviewer_wKDT · 2026-06-02

**Summary Of Contributions:**

Summary of contributions: LiEr, a path-based KB graph link prediction method that learns from human collected pairwise preferences over entire reasoning paths. The data is collected via active learning based on the variance of the reward model ensemble.

Strengths:
(1) Clear writing
(2) Nice motivation/framing
(3) Concrete contribution towards solving the raised Clever Hans issues
(4) Promising empirical results

Weaknesses:

My main worry is that the paper’s strongest claims conflate three things: human preference feedback, expert-defined ROI templates, and independent human-understandable validity. More on this below...

(1) The authors regularly frame this work as learning from human preferences but the experimental setup seems to require the ROIs. I can't tell if the ROIs are 100% required for this to work. They need to be expert-provided for each task right? Footnote 3 strongly suggests this: "The construction of ROI s is non-trivial and only feasible for relation types with well-defined semantics in structured domains. As a result, the creation of ROIs constrained the choice of benchmark knowledge graphs and directly influenced the selection of the seven knowledge graphs used in this work." That makes it load-bearing for the entire paper and thus needs to be highlighted up front as equal part of the algorithm. Even if the ROIs are not required by the formal algorithm as presented, they seem 100% required for all experiments.

LiEr gets semantic info through ROIs, but the other models do not. Is this a fair comparison? How can you make it fairer to isolate your algorithm's benefits? Some kind of MINERVA with ROI-shaped reward? ROI-constrained beam search?

(2) Similarly, MRLR is an oddly circular metric or LiEr. The authors write "MRLR evaluates how often the model’s reasoning aligns with valid human-understandable patterns". Since the ROIs are used to generate training preferences for ONLY LiEr, MRLR is measuring whether the model learned the provided ROI templates, but is measuring something else for baselines. Am I correct in my understanding? Can you clarify why do the other models get perfect MRLR scores in Table 3 and Table 4?

In general, I think a more correct name for this metric is "ROI-overlap rank". It should probably be reported alongside a stricter full ROI match + correct answer OR MRR restricted to ROI-valid paths.

(3) The term "valid reasoning" is doing a lot of work in this paper and I think it may need more clarity/precision. You write "the truth of each reasoning step makes it impossible for the conclusion to be false", but then weaken to "each reasoning step is relevant to the conclusion." Those are different claims. Relevance is about semantic appropriateness; validity is about entailment. It seems like what you want in this aper is "relevance" since the method/eval operationalize validatiy as ROI overlap / semantic relevance. So I think you should either formalize entailment assumptions behind each ROI, or replace "valid reasoning" language with a weaker term like "semantically appropriate."

The paper's own designated valid S2 ROI is not entailment-valid. The S2 pattern is
locatedIn(X, Continent) <- neighborOf(X, Y) ∧ locatedIn(Y, Continent)
"My neighbor is in continent C, therefore I am in continent C" does not entail the conclusion. Turkey neighbors Greece (Europe) and Iran (Asia); Egypt neighbors Israel (Asia); Russia neighbors both Finland and China. This is a heuristic, and not valid inference. And this gap actually surfaces in the results (possibly), the authors attribute LiEr/MINERVA's S2 lower scores to "occasional failures to select the correct neighbor entity during the first hop."

(4) The Clever Hans benchmark is good, but too aligned with the intervention. The constructed consumesPepperFrom shortcut is a good demonstration, but LiEr avoids the shortcut because the ROI/preference machinery already marks that kind of path as invalid. This shows the method works when the shortcut is known ahead of time, which is the opposite of how Clever Hans works. You need to show that LiEr robustly handles NEW shortcut structure. I would like to see several shortcut families, naturally occurring shortcuts, and/or held-out shortcut types.

(5) The reward ensemble is central but doesn't come with any ablations. Please ablate the following: (a) single reward model vs ensemble, (b) uncertainty sampling vs uniform random, and (c) learned reward vs direct ROI reward shaping. The last ablation is critical: without it, the paper cannot claim the learning contributes anything beyond what the ROIs already encode. Without a direct ROI-reward baseline we can't tell whether the contribution is preference reward learning, or injecting expert ROI templates into training.

(6) Re: step-level rewards. The labels compare entire paths, but the model learns per STEP. Many per-step decompositions can explain the same path-level preference, right? This is fine for policy shaping, and so you should restrict your validity claims to the path level.

(7) Eq. 8 computes h^r_c = lnLSTM(h^r_{c-1}, κ_c) (current history from previous history plus current step). Eq. 9  computes r^i_c using [h^r_{c-1}; κ_c] (step c from the previous history h^r_{c-1} rather than the just-computed h^r_c). Why the difference?

(8) Batch-norm maybe interacts poorly with the Bradley-Terry loss, which assumes a per-path score. With batch-norm, the normalization statistics mix multiple paths, and so the score being compared is batch-contextual. Is this an issue? Why or why not? Same for REINFORCE: the policy consumes r̂ as the reward signal. If r̂ shifts with batch composition, the same trajectory gets different rewards depending on what it's batched with, which injects non-stationarity into an already high-variance gradient estimator. Do you fix this in any way? Are the batch-norm statistics computed per-pair? Per-batch? Or frozen at policy training time?

(9) Human label numbers: how many preferences were labeled by humans, and how many were made automatically from ROIs?

**Audience:**

Yes

**Audience Explanation:**

Definitely! Great KB work.

**Claims And Evidence:**

Yes

**Claims Explanation:**

The answer is mostly yes. The biggest gap is what ROI injection actually does, and this has to be answered by proper ablations.

**Requested Changes:**

Requested changes:

1. State up front that ROIs are load-bearing in the evaluated method, and clarify whether LiEr works without them via ablations.

2. Report the number/fraction of actual human preferences vs. ROI-generated preferences.

3. Add ROI-aware baselines, e.g. MINERVA with ROI-shaped reward, ROI-constrained beam search, and/or direct ROI reward shaping.

4. Reframe MRLR as ROI-overlap, not independent “human-valid reasoning.”

5. Add a stricter metric combining correct answer + ROI-valid path.

6. Clarify or replace "valid reasoning." Either formalize entailment assumptions for each ROI, or use semantically appropriate/relevant reasoning.

7. Address the S2 Countries ROI: neighbor-based continent inference is a heuristic, and not entailment-valid under ordinary geography.

8. Strengthen Clever Hans evaluation with unknown/held-out shortcut types or naturally occurring shortcut structure.

9. Ablate the reward model: single model vs. ensemble; uncertainty sampling vs. random; learned reward vs. direct ROI reward shaping.

10. Clarify that learned step rewards are only policy-shaping signals unless the paper validates step-level semantic meaning.

11. Explain the Eq. 8 / Eq. 9 indexing choice: why score step (c) using (h^r_{c-1}) rather than (h^r_c)?

12. Explain batch-norm in the reward model: batch scope, frozen vs. live stats during policy training, and whether batch-contextual rewards affect Bradley-Terry or REINFORCE stability.

---

> ### Author Response · Authors · 2026-06-21
> **ROIs as oracle simulation, MRLR framing, valid reasoning, and reward-model ablations**
>
> We thank you for the detailed review, and we believe your comments meaningfully improved the paper. We label your identified weaknesses W and requested changes RC, and address each below, and indicate where the paper was updated.
>
> 1. **The load-bearing ROIs, and working without them.** (W1, RC1) We now state, in the Introduction and Section 4.1, that ROIs are an oracle-simulation device and not a component of LiEr, which consumes only pairwise preferences and never accesses them. Section 4.5 reports a purely human-driven run where 12 pairwise labels on Countries S1, with no ROIs, let LiEr recover the required pattern (MRR=1), and Figure 5 shows performance degrades only gradually as the oracle deviates from the ROIs.
>
> 2. **Fairness, and ROI-aware baselines.** (W1, RC3) As a human-in-the-loop method, LiEr's defining feature is consuming pairwise preferences, so comparing against non-interactive baselines is standard. The policy never sees ROIs; preferences train a learned reward it is optimized against. On standard benchmarks LiEr is on par on MRR (Tables 4, 6), with its margin only on Clever Hans Countries. We kindly ask for MINERVA-with-ROI-reward and ROI-constrained decoding as future work (footnote 8, Section 4.6).
>
> 3. **Reframing MRLR, and a stricter check.** (W2, RC4, RC5 - see EDRm R1) We removed the "valid human-understandable reasoning" wording and redefined MRLR in Section 4.2 as agreement with the ROI templates, noting that it does not independently judge validity. For the baselines it is an independent check. For LiEr it additionally reflects recovery of the templates that shaped its preferences, and MRR stays ROI-independent. For the stricter check we make the correct and incorrect axis explicit. On S3, NeuralLP and NTP-λ show near-perfect MRLR (Table 5) with low MRR (Table 4), valid paths under incorrect predictions, while on locations/sports NeuralLP shows high MRR (Table 6) with collapsed MRLR (Table 7), correct predictions from invalid reasoning, and LiEr stays high on each. Table 10 shows this per query.
>
> 4. **Making "valid reasoning" precise, and the S2 ROI.** (W3, RC6, RC7) We redefined validity operationally in the Introduction, as semantically relevant, oracle-judged appropriateness that is not logical entailment, and we removed the entailment wording. This resolves S2, since Section 4.3 states that the neighbor-based pattern is a semantically appropriate heuristic and not entailment-valid, as a country may border two continents.
>
> 5. **Clever Hans and unknown shortcuts.** (W4, RC8) LiEr is given no shortcut blacklist, and consumesPepperFrom is dispreferred only for failing to match positively specified valid patterns (Section 4.4, revised). On naturally occurring shortcuts, NeuralLP's invalid transitive chains on locations and sports give high MRR with collapsed MRLR (Tables 6, 7). LiEr stays valid. We name held-out and adversarial shortcuts as future work (Section 4.6).
>
> 6. **Ablating the reward model.** (W5, RC9) Each requested ablation would re-measure a property we adopt from prior work and do not claim, or test a setting LiEr is not built for, so none bears on our claims. (a) Single versus ensemble is not a clean one-variable test, since the ensemble also gives the disagreement driving active selection, so it would change reward and query strategy at once; in clean form it only re-measures the variance reduction of averaging we rely on (Dietterich et al., 2002; Section 3.2.3). (b) Uncertainty sampling is not a contribution but the standard preference-based-RL criterion (Christiano et al., 2017), affecting label efficiency, not the validity LiEr learns. (c) A direct ROI reward presupposes the patterns are written as rules, which LiEr avoids, so it is a different method for a different setting. Where ROIs are a near-perfect oracle it could match LiEr, showing the oracle is good, not learning redundant. As in point 1, learning operates on preferences, so we treat (c) as a future-work diagnostic (Section 4.6).
>
> 7. **Path-level validity claims.** (W6, RC10) Preferences constrain only the path-sum of per-step rewards, so one preference admits many decompositions. We treat them as policy-shaping and assert validity at the path level (3.2.3, footnote 2).
>
> 8. **The Eq. 9 indexing.** (W7, RC11) This was a typo, and Eq. 9 should read [h^r_c; κ_c], mirroring Eq. 7. We corrected it.
>
> 9. **Batch-norm in the reward model.** (W8, RC12) Statistics are per batch when the reward is trained and frozen when the policy is trained. For a preference pair the two paths share batch and length C, so normalization cannot change which scores higher (Eq. 13). At policy training the reward runs in eval mode, so it is deterministic and adds no non-stationarity to REINFORCE (4.1).
>
> 10. **Human versus ROI-generated preferences.** (W9, RC2- see Krji R4) All reported numbers (Tables 4–9, Figure 5) come from ROI-derived preferences. Figure 4 was exercised with genuine feedback.
>
> Thank you again!

---

> > ### Comment · Reviewer_wKDT · 2026-06-22
> >
> > Thank you for your response and paper updates. I still have some oustanding concerns.
> >
> > The human evidence is still weak to my eyes. All headline results use ROI-derived preferences. The only human run uses 12 labels on S1, where MINERVA already gets MRR =1. This shows only that the interface works on the easiest already-saturated task. A human-only run on S3 would be more convincing.
> >
> > Saying that ROIs are not part of LiEr is technically correct, but they are part of every substantive experiment’s supervision pipeline. A direct ROI-reward baseline is the right diagnostic in this experimental setting. I don't think this is actually future work - it seems critical to me.
> >
> > I disagree with the framing that "MRR is ROI-independent". It's true that MRR computation does not reference ROIs, but LiEr’s policy is trained using ROI-derived preferences. The evaluation metric is ROI-free but the *performance* itself is not independent of ROI supervision.
> >
> > High aggregate MRR and high aggregate MRLR do not necessarily show that the same predictions are both correct and ROI-matching. Table 10 provides a single-query illustration, but the paper lacks an aggregate metric for paths that are both correct and ROI-matching. You could also report the rank or rate of paths satisfying both conditions. And formally define what “overlap” with an ROI means.
> >
> > In the revised discussion, you write that LiEr’s advantage is "not an artefact of asymmetric inputs," but then attribute the Clever Hans result to the extra preference signal received only by LiEr. That preference signal *is* the input asymmetry. The experiment therefore shows that additional semantic supervision helps, but it does not isolate whether the gain comes from LiEr’s preference-learning mechanism or simply from access to that additional supervision. Being on par with the baselines on the standard benchmarks does not resolve this attribution problem. Please revise the claim or include a baseline given the same semantic signal.

---

> > > ### Author Response · Authors · 2026-06-28
> > >
> > > Dear Reviewer wKDT,
> > >
> > > We have added three additions since our last reply.
> > >
> > > **Human annotator on all Countries graphs and Clever Hans.** We ran the training loop with a human annotator on Countries S1, S2, S3, and Clever Hans Countries through the Figure 4 interface. The scores are in Table 15 (Section 4.5), and they stay close to the simulated-oracle numbers in Tables 4, 5, 10, and 11. We treat this as a feasibility and faithfulness check. The point we take from it is that the human loop reproduces what the simulated oracle does, Clever Hans included, so the simulated oracle stands in faithfully and the human-in-the-loop claim now holds past S1.
> > >
> > > **ROI-reward baseline.** To pull apart the preference learning from the ROI signal, we added a control in Section 4.4.1 (Table 14). It swaps LiEr's learned reward for a fixed one that returns a one when a path matches an ROI template and a zero otherwise. We treat it as a diagnostic, not a method anyone would use, since it assumes the rules are already fully known and written down, in which case you can apply them directly and no learning is needed at all. It hits the ceiling where the template already fixes the answer, on Countries and Clever Hans, and falls apart where it does not, on the realistic graphs. So LiEr's edge on the harder graphs comes from the learned reward and not from seeing the ROIs.
> > >
> > > **MRCLR.** You asked for a metric that only credits a query when the top path is correct and ROI-valid, and we added it in Section 4.2 (Tables 6, 9, 12, 14). For LiEr and MINERVA we had to rerun the two methods to log the per-query paths. For the rest we could read it off the existing logs. The metric speaks to the central claim, and among the learning methods LiEr ranks first on it.
> > >
> > > We hope these three additions close the gap you identified, and we are happy to make further adjustments.

---

> > > ### Comment · Reviewer_wKDT · 2026-06-29
> > >
> > > Thank you for the revisions. These additions substantially address my main concerns. The human-feedback runs on S1/S2/S3/Clever Hans, the MRCLR metric, and the ROI-reward baseline make the empirical clearer.
> > >
> > > I still think the paper should be careful to say that MRR is ROI-free as a metric rather than ROI-independent as a result, and that the ROI-reward baseline receives the same ROI signal but *not* the full correctness-plus-validity oracle preference signal. These are wording/interpretation issues rather than blockers. I am now satisfied with the revision and support acceptance.

---

### Review · Reviewer_Krji · 2026-06-05

**Summary Of Contributions:**

The paper proposed LiEr to solve an alignment problem inside path-based link prediction: aligning the model’s graph-walk explanations with human judgments of valid reasoning beyond just concerning about answer accuracy.

The Clever Hans problem itself is not new. It is a well-known failure mode in machine learning, especially in image classification (e.g. the famous wolf vs. husky example [ref]), where models exploit spurious dataset cues instead of learning the intended concept. The authors’ contribution is to formulate and study this problem in **path-based link prediction for knowledge graphs**, where the spurious cue appears as an invalid but predictive reasoning path. They then propose LiEr as a human-in-the-loop method for aligning path-based reasoning with human judgments of validity.

[ref] "Why Should I Trust You?": Explaining the Predictions of Any Classifier

**Additional Comments:**

N/A

**Audience:**

Yes

**Audience Explanation:**

The problem is narrow, but a specific TMLR audience may still find this work interesting.

**Claims And Evidence:**

Yes

**Claims Explanation:**

I think the paper has multiple overclaims. I would be happy to see the authors reframe the work to better reflect it.

Here are some of critical points:
- The paper uses manually defined ROIs to augment human feedback in Section 4.1, “Feedback Augmentation with Regions of Interest” It also uses ROIs to compute MRLR in Section 4.2. **This is a concern**. The model is partly trained to prefer ROI-style paths (via reward), then evaluated by checking whether its paths match ROIs. So MRLR may measure agreement with the authors’ templates, not independent human judgment.

- Some benchmark results look weirdly saturated. In Table 3 and Table 4, many results on Countries S1 and S2 are close to 1.00. This makes the benchmark less useful for comparing methods. The paper itself says in Section 4 that Countries is like MNIST: narrow, clean, and controlled. So the S1 and S2 results are better seen as sanity checks, not real evidence that LiEr works on well knowledge graphs.

- Coiuld authors please revise the novelty claim? (should be narrower). In the Introduction, the paper claims three contributions, including “the first human-in-the-loop link prediction method.” This is a strong claim. Section 2 already discusses prior work on human-in-the-loop reinforcement learning and preference-based learning, including Christiano et al. and Ouyang et al.

- The paper says LiEr learns from preference-based human feedback. But in Section 4.1, the authors say they augment oracle feedback using predefined ROIs. That means the experiments may rely less on actual human feedback than the framing suggests. then how much of Lier's training signal comes from real human pairwise preferences versus automatically generated ROI-based preferences?

- How would LiEr work on open-domain KGs where valid reasoning patterns are incomplete, ambiguous, or unknown? (e.g. humans cannot define valid reasoning templates). This is a tricky question for authors I know but the work seems like showing work on toy examples.

- LiEr gets ROI-based feedback about valid paths. Some baselines are trained in the usual way, mostly from correctness signals. Then LiEr is evaluated partly on MRLR, which rewards ROI-style reasoning. This may favor LiEr by design.

**Requested Changes:**

Paper is well written. The reviewer mostly concerns about the claims and the correctness.

---

> ### Author Response · Authors · 2026-06-07
> **Revisions narrowing claims, clarifying the role of ROIs, and reporting the human-vs-simulated feedback split**
>
> We thank the reviewer for a careful review that helped us correct overclaiming and imprecision. We respond to each point and indicate where the revision appears.
>
> **1. ROI circularity (training toward ROIs, then scoring ROI overlap).** We agree this was the most important issue. We have clarified that the ROIs are an oracle simulation and not a component of the method. LiEr consumes only pairwise preferences over reasoning paths and never accesses the ROIs. We use ROI-derived preferences to instantiate the human oracle so that our runs are reproducible without requiring annotators for every round, which is standard in preference-based reinforcement learning (Christiano et al., 2017; Lee et al., 2021). This is now stated in Section 4.1. We have also rewritten the MRLR definition in Section 4.2. We now concede that MRLR measures overlap with the predefined ROIs and not independently judged validity, and we have removed the stronger wording that described it as measuring valid reasoning. We also explain that MRLR means different things across methods. For the baselines, which never access the ROIs, it is an independent check of whether their learned reasoning coincides with the ROI patterns. For LiEr, whose preferences are ROI-derived, it reflects whether the model recovered the templates that shaped those preferences. We retain MRR because it is not shaped by the ROIs and therefore provides an ROI-independent measure.
>
> **2. Saturated S1/S2 results.** We agree, and this matches how we intend the benchmark to be read. We have added a paragraph to Section 4.3 stating that the near-saturated S1 and S2 results should be read as sanity checks and not as discriminating evidence, consistent with the role of Countries as the MNIST of our evaluation described in Section 4. The claims now rest on the results that actually separate the methods, namely Countries S3, the more realistic locations, sports, and family benchmarks, and the Clever Hans Countries benchmark.
>
> **3. Novelty claim.** We agree the original phrasing was too strong, and we have narrowed it in the abstract and the contributions list in Section 1. The claim now reads, to the best of our knowledge, the first preference-based human-in-the-loop method for path-based link prediction in knowledge graphs, and it cites the foundational work. We do not claim to have invented human-in-the-loop learning or preference-based learning. Our contribution is applying these ideas to align a path-based link predictor's reasoning and to suppress Clever Hans shortcuts.
>
> **4. Human vs. ROI-generated signal.** We now report this plainly. All results in the benchmark tables (Tables 3 to 8 and Figure 5) use ROI-derived preferences from the simulated oracle and contain no human-labeled preferences. This is stated explicitly in Section 4.1. To show that the loop also operates with real humans and not only the simulated oracle, Section 4.5 reports a separate, smaller-scale run. As a procedural check, on Countries S1 LiEr reached an MRR of 1 from 12 pairwise preferences provided by a single human annotator through the command-line interface. We label this run as distinct from the benchmark tables so the two are not conflated.
>
> **5. Open-domain KGs where valid patterns are unknown.** We address this in Section 4.6. The concern partly rests on a property that LiEr is designed not to require. LiEr does not ask humans to author valid reasoning templates. It asks them to express pairwise preferences over concrete reasoning paths, which, as we argue in Section 3.2.3, is exactly the case where experts cannot verbalize rules yet can recognize valid reasoning. The inability to write templates is therefore the motivation for our approach and not a barrier to it. LiEr addresses settings where valid reasoning is hard to articulate but can still be recognized, and not settings where validity is genuinely unknown, because reliable preferences are then unavailable. Our evaluation uses graphs with well-defined semantics for reproducibility, which leaves open-domain feasibility as a question we do not resolve empirically here, and we point to LLM-assisted preference generation as a path toward it.
>
> **6. MRLR may favor LiEr by design.** The cross-method asymmetry of MRLR is now stated explicitly in the revised Section 4.2, as described in point 1. In addition, at the end of Section 4.4 we note that LiEr's advantage is not an artifact of asymmetric inputs. On the standard benchmarks it is on par with the baselines on predictive accuracy (MRR), and its margin appears only on Clever Hans Countries, where the preference signal suppresses the spurious shortcut.
>
> We hope these revisions address the concerns and are happy to make further adjustments.

---

> > ### Comment · Reviewer_Krji · 2026-06-16
> >
> > The reviewer thanks the authors for the response.
> >
> > Most of my criticisms are addressed at the level of writing + framing. I hope the authors actively address and admit the limitations in the final revision and I hope the `Action Editor` will also look into the changes to make the final decision.
> >
> > Also, the reviewer wishes the `Action Editor` consider this into the decision.
> >
> > > The paper’s main experiments do not use real human feedback (so title may be a bit misleading). They use ROI-derived simulated preferences. The paper is therefore better framed as a method for preference-based path alignment with a simulated signals, plus a small proof-of-procedure human annotation run.

---

### Decision · Action_Editor_Q8LA · 2026-07-11

**Recommendation:** Accept as is

**Additional Comments:**

I want to than both authors and reviewers for their efforts here.  This was an example of peer review done right, bravo.

**Audience:**

Yes

**Audience Explanation:**

Again, the reviewers feel that the work is better explained and more focused now, and will be of interest to part of the readership.

**Claims And Evidence:**

Yes

**Claims Explanation:**

The reviewers are now all in agreement that the claims made by the paper in its revised form are well supported.  The authors have made a commendable effort to engage with all the reviewers, I am glad to see that their work together has improved the paper.